# Kondo blockade due to quantum interference in single-molecule junctions

Andrew K. Mitchell[1,2], Kim G.L. Pedersen[3], Per Hedegård[4] & Jens Paaske[5]

Molecular electronics offers unique scientific and technological possibilities, resulting from both the nanometre scale of the devices and their reproducible chemical complexity. Two fundamental yet different effects, with no classical analogue, have been demonstrated experimentally in single-molecule junctions: quantum interference due to competing electron transport pathways, and the Kondo effect due to entanglement from strong electronic interactions. Here we unify these phenomena, showing that transport through a spin-degenerate molecule can be either enhanced or blocked by Kondo correlations, depending on molecular structure, contacting geometry and applied gate voltages. An exact framework is developed, in terms of which the quantum interference properties of interacting molecular junctions can be systematically studied and understood. We prove that an exact Kondo-mediated conductance node results from destructive interference in exchange-cotunneling. Nonstandard temperature dependences and gate-tunable conductance peaks/nodes are demonstrated for prototypical molecular junctions, illustrating the intricate interplay of quantum effects beyond the single-orbital paradigm.

[1] School of Physics, University College Dublin, Dublin 4, Ireland. [2] Institute for Theoretical Physics, Utrecht University, Princetonplein 5, Utrecht 3584 CE, The Netherlands. [3] Institut für Theorie der Statistischen Physik, RWTH Aachen University, Aachen 52074, Germany. [4] Niels Bohr Institute, University of Copenhagen, Copenhagen DK-2100, Denmark. [5] Center for Quantum Devices, Niels Bohr Institute, University of Copenhagen, Universitetsparken 5, Copenhagen DK-2100, Denmark. Correspondence and requests for materials should be addressed to A.K.M. (email: Andrew.Mitchell@ucd.ie).

Perhaps the most important feature of nanoscale devices built from single molecules is the potential to exploit exotic quantum mechanical effects that have no classical analogue. A prominent example is quantum interference (QI), which has already been demonstrated in a number of different molecular devices[1–9]. QI manifests as strong variations in the conductance with changes in molecular conformation, contacting or conjugation pathways or simply by tuning the back-gate voltage in a three-terminal setup. Another famous quantum phenomenon, relevant to single-molecule junctions with a spin-degenerate ground state, is the Kondo effect[10–16], which gives rise to a dramatic conductance enhancement below a characteristic Kondo temperature, $T_K$. Strong electronic interactions in the molecule cause it to bind strongly to a large Kondo cloud[17,18] of conduction electrons when contacted to source and drain leads. A hallmark of the Kondo effect is the proliferation of spin flips as electrons tunnel coherently through the molecule, ultimately screening its spin by formation of a many-body singlet[19]. In this article, we uncover the intricate interplay of these two quantum effects, finding that the combined effect of QI and Kondo physics has highly non-trivial consequences for conductance through single-molecule junctions, and can even lead to an entirely new phenomenon—the Kondo blockade.

The Kondo effect is also routinely observed in semiconductor and nanotube quantum dot devices[20–24], which are regarded as lead-coupled artificial atoms[25] and as such are often well described in terms of a single active interacting quantum orbital, tunnel-coupled to a single channel of conduction electrons comprising both source and drain leads. This Anderson impurity model (AIM) is by now rather well understood[19], and a quantitative description of the Kondo peak in single quantum dots can be achieved within linear response using non-perturbative methods such as the numerical renormalization group (NRG)[26,27]. In particular, the conductance is a universal function of $T/T_K$, meaning that data for different systems collapse to the same curve when rescaled in terms of their respective Kondo temperatures[20–23]. Indeed, even for multi-orbital molecular junctions, experimental conductance lineshapes have in some cases been successfully fit to the theoretical form of the AIM, suggesting that an effective single-orbital, single-channel description is valid at low temperatures[15,16]. However, some molecular junctions[14,28,29] apparently manifest nonuniversal behaviour and unconventional gate voltage dependences of conductance and $T_K$, hinting at new physics beyond the standard single-orbital paradigm.

The breakdown of the AIM is well-known in the context of coupled quantum dot devices[30–44], which can be viewed as simple artificial molecular junctions due to their multi-orbital structure and the coupling to distinct source and drain channels. Already the extension to two or three orbital systems has lead to the discovery of striking phenomena such as the ferromagnetic Kondo effect[31–33] corresponding to a sign change of the exchange coupling, and multistage[35,36] or frustrated[37–44] screening.

In the following, we argue that a similar kind of multi-channel, multi-orbital Kondo physics accounts for the behaviour of real molecular junctions, and can be understood as a many-body QI effect characteristic of the orbital complexity and strong electronic correlations in molecules. On entirely general grounds, we construct an effective model describing off-resonant conductance through single-molecule junctions with a spin-degenerate ground state, taking into account both interactions leading to Kondo physics, and orbital structure leading to QI. The physics of this generalized two-channel Kondo (2CK) model (including both potential scattering and exchange-cotunneling) is discussed in relation to the local density of states and observable conductance. We demonstrate how renormalized Kondo resonant conductance evolves into a novel Kondo blockade regime of suppressed

conductance due to QI (Fig. 1). As an illustration, we consider two simple relevant molecular examples, whose properties can be tuned between these limits using gate voltages to provide functionality as an efficient QI-effect transistor.

## Results

**Models and mappings.** The Hamiltonian describing single-molecule junctions can be decomposed as,

$$H = H_{mol} + H_g + H_{leads} + H_{hyb}. \qquad (1)$$

Here $H_{mol}$ describes the isolated molecule, and contains all information about its electronic structure and chemistry. The first-principles characterization of molecules is itself a formidable problem when electron–electron interactions are taken into account. In practice however, the relevant molecular degrees of freedom associated with electronic transport are often effectively decoupled. This is the case for many conjugated organic molecules, where the extended $\pi$ system can be treated separately in terms of an extended Hubbard model[45]. Reduced multi-orbital models have also been formulated using *ab initio* methods[46–49].

The leads are modelled as non-interacting conduction electrons with $H_{leads} = \sum_{\alpha\sigma k} \epsilon_k c^\dagger_{\alpha\sigma k} c_{\alpha\sigma k}$ where $c^\dagger_{\alpha\sigma k}$ creates an electron in lead $\alpha = $ s, d (source, drain) with momentum (or other orbital quantum number) $k$, and spin $\sigma = \uparrow, \downarrow$. The dispersion $\epsilon_k$ corresponds approximately to a flat density of states $\rho(E) = \rho_0 \theta(D - |E|)$, inside a band of width $2D$.

The molecule is coupled to the leads via $H_{hyb} = \sum_{\alpha\sigma} (t_\alpha d^\dagger_{i_\alpha\sigma} c_{\alpha\sigma} + \text{H.c.})$, where $c_{\alpha\sigma} = t_\alpha^{-1} \sum_k t_{\alpha k} c_{\alpha\sigma k}$ is the localized orbital in lead $\alpha$ at the junction, and $d_{i_\alpha\sigma}$ is a specific frontier orbital $i_\alpha$ of the molecule, determined by the contacting geometry. The molecule-lead hybridization is local, and specified by $\Gamma_\alpha = \pi\rho_0 |t_\alpha|^2$.

The number of electrons on the molecule, $\mathcal{N} = \langle \sum_{i\sigma} d^\dagger_{i\sigma} d_{i\sigma} \rangle$, is controlled by a gate voltage, incorporated in the model by $H_g = -eV_g \sum_{i\sigma} d^\dagger_{i\sigma} d_{i\sigma}$ which shifts the energy of all molecular orbitals. Deep inside the Coulomb diamond[10], a substantial charging energy, $E_C$, must be overcome to either add or remove electrons from the molecule. Provided $\Gamma_{s,d} \ll E_C$ the Hamiltonian (1) can therefore be projected onto the subspace with a fixed number of electrons on the molecule. In general this requires full diagonalization of the isolated $H_{mol}$ in the

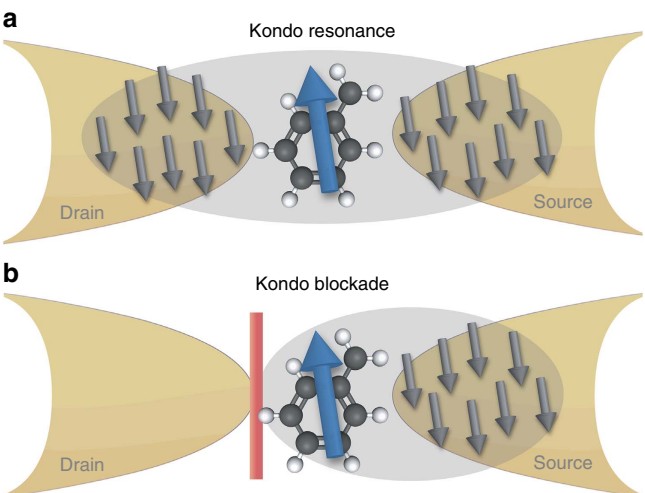

**Figure 1 | Interplay between quantum interference and electronic interactions in single molecule junctions.** (**a**) Enhanced Kondo resonant conductance; (**b**) Kondo blockade, where conductance precisely vanishes. Tuning between **a**,**b** by applying a back-gate voltage allows efficient manipulation of the tunnelling current.

many-particle basis. Charging a neutral/spinless molecule by applying a back-gate voltage to add or remove an electron typically yields a net spin-$\frac{1}{2}$ state. For odd-integer $\mathcal{N}$, we therefore assume that the molecule hosts a spin-$\frac{1}{2}$ degree of freedom, **S**. Higher-spin molecules can arise, but are not considered here (the generalization is straightforward). To second-order in the molecule-lead coupling $H_{hyb}$, we then obtain[50] an effective model of generalized 2CK form, $H_{2CK} = H_{leads} + H_{ex}$, where

$$H_{ex} = \sum_{\alpha\alpha'\sigma\sigma'} \left( \tfrac{1}{2} J_{\alpha\alpha'} \mathbf{S} \cdot \boldsymbol{\tau}_{\sigma\sigma'} + W_{\alpha\alpha'} \delta_{\sigma\sigma'} \right) c^\dagger_{\alpha\sigma} c_{\alpha'\sigma'}. \quad (2)$$

Here $\boldsymbol{\tau}$ denotes the vector of Pauli matrices. The form of $H_{ex}$ is guaranteed by spin-rotation invariance and Hermiticity. Further details of the 2CK mapping are provided in Methods. The cotunneling amplitudes form matrices in source-drain space,

$$J = \begin{bmatrix} J_{ss} & J_{sd} \\ J_{sd} & J_{dd} \end{bmatrix} \quad W = \begin{bmatrix} W_{ss} & W_{sd} \\ W_{sd} & W_{dd} \end{bmatrix}, \quad (3)$$

and are referred to as respectively exchange, and potential scattering terms. These 2CK parameters depend on the specifics of molecular structure and contacting geometry in a complicated way, and must be derived from first-principles calculations for the isolated molecule. This generalized 2CK model hosts a rich range of physics; the non-Fermi liquid critical point[38] is merely a single point in its parameter space. Furthermore, any conducting molecular junction must have a Fermi liquid ground state, as demonstrated below.

Any off-resonant molecule hosting a net spin-$\frac{1}{2}$ is described by the above generalized 2CK model at low temperatures $T \ll E_C$. The physics is robust due to the large charging energy deep in a Coulomb diamond (charge fluctuations only dominate at the very edge of the Coulomb diamond[29]). In fact, the physics of the effective model can be regarded as exact in the renormalization group (RG) sense, despite the perturbative derivation of equation (2). Corrections to $H_{ex}$ obtained in higher-order perturbation theory are formally RG irrelevant, and can be safely neglected because they get smaller and asymptotically vanish on decreasing the temperature. They cannot affect the underlying physics; only the emergent energy scales can be modified (this effect is also small, since the corrections are suppressed by $E_C$).

Experimentally relevant physical observables such as conductance can therefore be accurately extracted from the solution of the effective 2CK model (Methods). This requires sophisticated many-body techniques such as NRG, which theoretically 'attach' the source and drain leads non-perturbatively[26,27]. All microscopic details of a real molecular junction are encoded in the 2CK parameters $J$ and $W$, which serve as input for the NRG calculations. In particular, destructive QI produces nodes (zeros) in these parameters. Furthermore, QI nodes can be simply accessed by tuning the back-gate voltage $V_g$, as was shown recently in ref. 45 for the case of conjugated organic molecules.

Importantly, two different types of QI can arise in molecular junctions due to the electronic interactions. The QI can either be of standard potential scattering type (zeros in elements of $W$) or of exchange type (zeros in elements of $J$). Potential scattering QI is analogous to that observed in non-interacting systems described by molecular orbitals. For interacting systems such as molecular junctions (which typically have large charging energies[10]), potential scattering QI can similarly be understood in terms of extended Feynman–Dyson orbitals, which are the generalization of molecular orbitals in the many-particle basis. Information on the real-space character of these orbitals, and how QI relates to molecular structure, can be extracted from the 2CK mapping. By contrast, exchange QI has no single-particle

analogue, and cannot arise in non-interacting systems. Indeed, interactions are a basic requirement for the molecule to host a spin-$\frac{1}{2}$ via Coulomb blockade. The spin wavefunction is again characterized by the Feynman–Dyson orbitals; depending on the molecule in question, the spin can be delocalized over the entire molecule.

In the following we uncover the effect of this QI on Kondo physics, highlighting two distinct scenarios for the resulting conductance—Kondo resonance and Kondo blockade. We then go on to show that this physics is indeed realized in simple examples of molecular junctions, and can be manipulated with gate voltages.

**Emergent decoupling**. The generalized 2CK model can be simplified by diagonalizing the exchange term in equation (2) via the unitary transformation $c_{\alpha'\sigma} = U_{\alpha'\alpha}\psi_{\alpha\sigma}$ such that

$$U^\dagger J U = \begin{bmatrix} J_e & 0 \\ 0 & J_o \end{bmatrix}, \quad J_{e/o} = J_+ \pm \delta \quad (4)$$

where $J_\pm = \frac{1}{2}(J_{ss} \pm J_{dd})$ and $\delta^2 = J_-^2 + J_{sd}^2$. Note that $W$ is not generally diagonalized by this transformation. The 'odd' channel decouples ($J_o = 0$) if and only if $J_{sd}^2 = J_{ss}J_{dd}$, as is the case when starting with a single-orbital Anderson model (see Supplementary Note 1). By contrast, real multi-orbital molecules couple to both even and odd channels (electronic propagation through the entire molecule yields $J_{sd}^2 \ll J_{ss}J_{dd}$ when off resonance).

However, electronic interactions play a key role here: the exchange couplings become renormalized as the temperature is reduced. A simple perturbative RG treatment hints at flow toward a two-channel strong-coupling state, since both $J_e$ and $J_o$ initially grow. But the true low-temperature physics is much more complex, as seen in Fig. 2 from the imaginary part of the scattering T-matrix $t_{\alpha\alpha}(\omega, T) = -\pi\rho_0 \text{Im} T_{\alpha\alpha}(\omega, T)$ obtained by NRG for the generalized 2CK model and plotted as a function of excitation energy $\omega$ at $T = 0$ (see Methods). The molecule spin is ultimately always Kondo-screened by conduction electrons in the more strongly coupled even channel since $J_e > J_o$ for any finite $\delta$. Indeed, any real molecular junction will inevitably have some degree of asymmetry in the source/drain coupling $J_-$, so that $\delta \geq J_-$ is always finite in practice. At particle–hole (ph) symmetry, the Friedel sum rule[19] then guarantees that $t_{ee}(0, 0) = 1$, characteristic of the Kondo effect. On the other hand, Kondo correlations with the less strongly coupled odd channel are cut off on the scale of $T_K$, and therefore $t_{oo}(0, 0) = 0$ (consistent with the optical theorem). These analytic predictions are verified by NRG results in the centre panels of Fig. 2.

In all cases the odd channel decouples on the lowest energy/temperature scales, and the problem becomes effectively single-channel. This is an emergent phenomenon driven by interactions, not a property of the bare model. Despite the emergent decoupling of the odd channel, the Kondo effect always involves conduction electrons in both source and drain leads for any finite $J_{sd}$. From the transformation defined in equation (4), the T-matrix in the physical basis can be expressed as $t_{\alpha\alpha}(\omega, T) = |U_{\alpha,e}|^2 t_{ee}(\omega, T) + |U_{\alpha,o}|^2 t_{oo}(\omega, T)$, such that $t_{\alpha\alpha}(0,0) = |U_{\alpha,e}|^2$ at ph symmetry—see left panels of Fig. 2.

Although the physics at $T = 0$ is effectively single-channel, the full temperature dependence is highly non-trivial due to the competing involvement of the odd channel (only for the oversimplified single-orbital AIM is the odd channel strictly decoupled for all $T$). The universal physics of the AIM is lost for $\delta = \frac{1}{2}(J_e - J_o) \neq J_+$ (or equivalently $J_{sd}^2 \neq J_{ss}J_{dd}$): conductance lineshapes no longer exhibit scaling collapse in terms of $T/T_K$. Indeed, Kondo screening by the even channel occurs on the scale $T_K^e \sim D \exp[-1/\rho_0 J_e]$, and hence depends on $\delta$. The

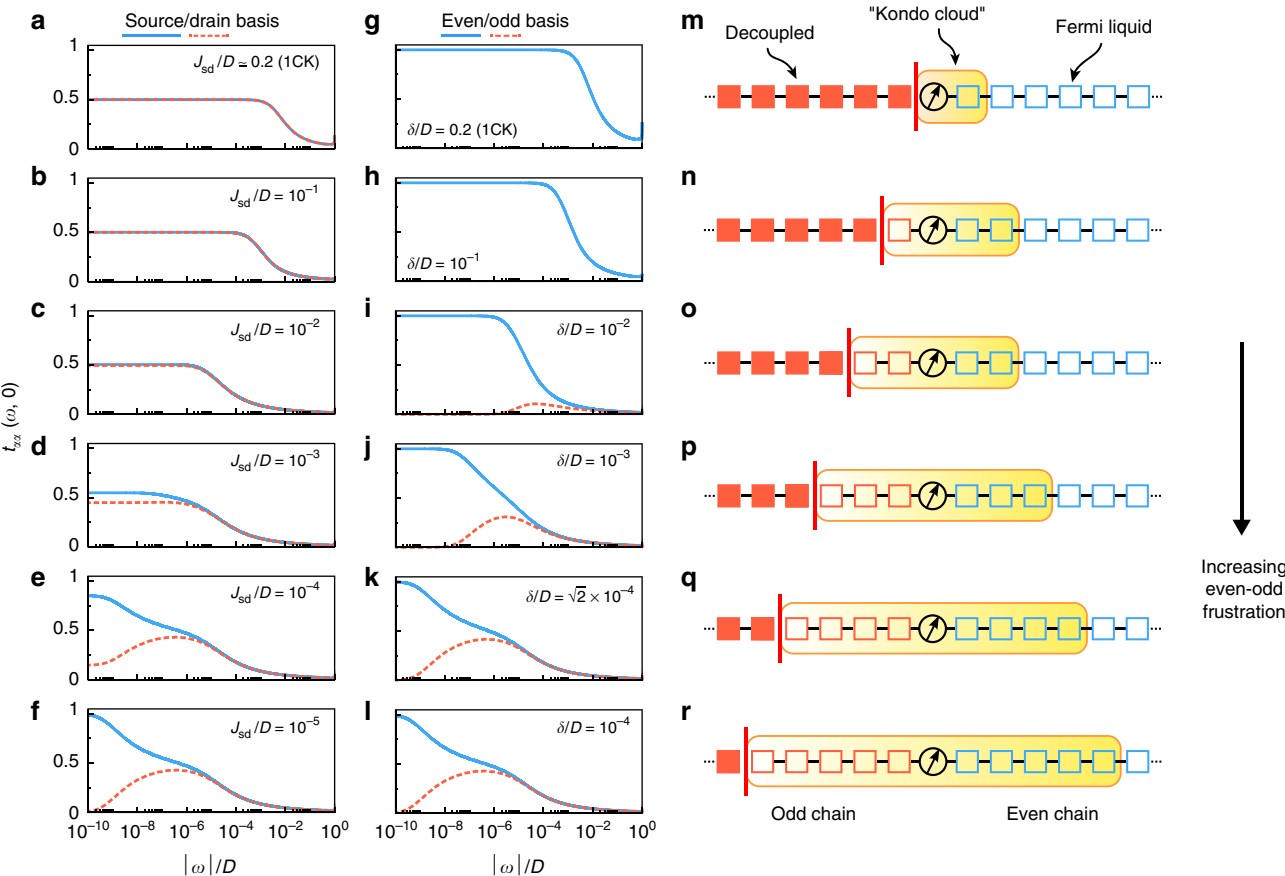

**Figure 2 | Decoupling and frustration due to the Kondo effect in molecular junctions. (a–f)** Imaginary part of the T-matrix, characterizing the effective energy-dependent exchange, in the physical basis of source (blue) and drain (red) leads at $T = 0$ for the effective 2CK model. $t_{\alpha\alpha}(\omega, 0)$ is related to the renormalized density of states in lead $\alpha$ at the junction. We take a representative molecule-lead coupling $J_+ = 0.2D$ with small but finite source/drain coupling asymmetry $J_- = 10^{-4}D$, and consider the effect of reducing the exchange-cotunneling $J_{sd}$ from **a–f**. Physically, this could be achieved by gate-tuning in the vicinity of a QI node. The frustration of Kondo screening is always eventually relieved on the lowest energy scales, below $T_{FL} \sim \min(T_K^e, T^*)$, because $\delta \geq J_-$ is always finite in any realistic setting. **(g–l)** Corresponding T-matrix in the even/odd (blue/red) lead basis. **(m–r)** Real-space competition between even/odd (blue/red) conduction electron channels, illustrated for the case where the leads are 1D quantum wires. The Kondo cloud (yellow) corresponds to the spatial region of high molecule-lead entanglement. For small $\delta \lesssim 10^{-3}D$ one has a 'Kondo frustration cloud' embodying incipient overscreening[17].

Kondo temperature itself can therefore acquire an unconventional gate voltage dependence, beyond the AIM paradigm.

For even smaller $\delta$, the Kondo effect occurs as a two-step process, with even and odd channels competing to screen the molecule spin. As in this case $J_e \approx J_o$, a frustration of Kondo screening sets in on the scale $T_K^e \approx T_K^o \equiv T_K^{2CK}$. The incipient frustration for $T \sim T_K^{2CK}$ results in only partial screening (the molecule is overscreened, producing non-Fermi liquid signatures[38,40–42,51]). The frustration is relieved on the much smaller scale[38] $T^* \sim D(\rho_0\delta)^2$. The even channel eventually 'wins' for $T \ll T^*$ and fully Kondo-screens the molecule spin, while the odd channel decouples. This dramatic breakdown of the single-orbital AIM paradigm is shown in Fig. 2, with the degree of even/odd frustration increasing from top to bottom. In practice, such frustration arises in a nearly symmetrical junction (small $J_-$), tuning in the vicinity of a QI node in $J_{sd}$ such that the perturbation strength $\delta$ is reduced. The first signatures of frustration appear in conductance when $T^* \lesssim T_K^e$. Only when $J_- = J_{sd} = 0$, such that $\delta = 0$, does the frustration persist down to $T = 0$; we do not consider this unrealistic scenario in the present work.

In real-space, the entanglement between the molecule and the leads is characterized by the Kondo cloud[18]—a large spatial region of extent $\xi_K \sim \hbar v_F / k_B T_K^e$ penetrating both source and drain leads ($v_F$ is the Fermi velocity). In the right panels of Fig. 2 we illustrate this for the case where the leads are 1D quantum wires; the real-space physics is then directly related to the T-matrix plotted in the left panels, as shown in ref. 17. Note that if the source/drain leads are 1D quantum wires, then the even/odd leads are also 1D quantum wires as depicted. For small $\delta$ (lower panels) we have instead a Kondo frustration cloud. The frustration is only relieved at longer length scales $\xi^* \sim \hbar v_F / k_B T^*$, beyond which the odd channel decouples.

**Conductance.** The current through a molecular junction is mediated by the cross terms coupling source and drain leads; the exchange and potential scattering terms $J_{sd}$ and $W_{sd}$ constitute two distinct conductance mechanisms. At high temperatures, the overall conductance can be understood from a simple leading-order perturbative treatment using Fermi's golden rule and is simply additive[45], $G/G_0 \sim (2\pi\rho_0)^2 [W_{sd}^2 + 3J_{sd}^2]$, with $G_0 = 2e^2h^{-1}$. However, at lower temperatures, electronic interactions lead to strong renormalization effects and rather surprising Kondo physics. Non-perturbative methods such as NRG must therefore be used to calculate the full temperature-

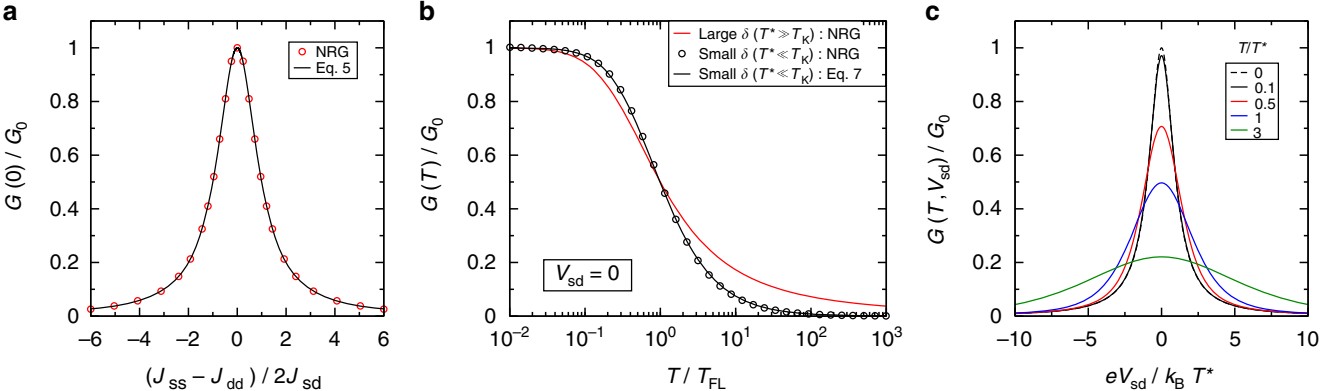

**Figure 3 | Kondo resonant conductance near a potential scattering quantum interference node.** (**a**) Zero-temperature linear conductance $G(0)$ as a function of derived 2CK parameters $J_{ss}$, $J_{dd}$ and $J_{sd}$ at $W_{ss} = W_{dd} = W_{sd} = 0$. (**b**) Limiting universal conductance curves $G(T/T_K)$ and $G(T/T^*)$ in the single-channel regime (large $\delta$, red line) and the frustrated two-channel regime (small $\delta$, black line and points), respectively. (**c**) Exact non-equilibrium conductance $G(T, V_{sd})$ as a function of bias voltage $V_{sd}$ at various temperatures in the frustrated regime of small $\delta$, from equation (7).

dependence of conductance, as described in Methods. Note that conductance through a molecular junction cannot be obtained simply from the T-matrix (except at $T = 0$).

The QI aspect of the problem is entirely encoded in the effective 2CK parameters, providing an enormous conceptual simplification. In particular, we identify two limiting QI scenarios relevant for conductance: $W_{sd} = 0$ or $J_{sd} = 0$. Exact analytic results, supported by NRG, show that the Kondo effect survives QI in the case of $W_{sd} = 0$ to give enhanced conductance at low temperatures (Figs 1a and 3), while a Kondo-mediated QI node in the total conductance is found for $J_{sd} = 0$, a Kondo blockade (Figs 1b and 4). We demonstrate explicitly that this remarkable interplay between QI and the Kondo effect arises in two simple conjugated organic molecules on tuning gate voltages in Fig. 5.

**Kondo resonance**. First we focus on conductance mediated exclusively by the exchange cotunneling term $J_{sd}$, tuning to a potential scattering QI node $W = 0$. Even though the bare $J_{sd}$ is typically small, it gets renormalized by the Kondo effect and becomes large at low temperatures. The Kondo effect therefore involves both source and drain leads (Fig. 2), leading to Kondo-enhanced conductance.

As shown in Supplementary Note 2, the fact that the odd channel decouples asymptotically implies the following exact result for the linear conductance,

$$G(T=0) = 4G_0 \sqrt{t_{ss}(0,0) t_{dd}(0,0)}$$
$$= G_0 \frac{4J_{sd}^2}{4J_{sd}^2 + (J_{ss} - J_{dd})^2}. \quad (5)$$

Note that any finite interlead coupling $J_{sd}$ yields unitarity conductance $G = G_0$ at $T = 0$ in the symmetric case $J_{ss} = J_{dd}$. The analytic result is confirmed by NRG in Fig. 3a, and further holds for all $T \ll T_K, T^*$. Equation (5) is an exact generalization of the standard single-orbital AIM result, $G(0)/G_0 = 4J_{ss}J_{dd}/(J_{ss} + J_{dd})^2 \equiv 4\Gamma_s\Gamma_d/(\Gamma_s + \Gamma_d)^2$, and reduces to it when $J_{sd}^2 = J_{ss}J_{dd}$.

The full temperature-dependence of conductance can also be studied with NRG. In all cases, we find Fermi-liquid behaviour $G(T) - G(0) \sim (T/T_{FL})^2$ at the lowest temperatures $T \ll T_{FL}$, with $T_{FL} = \min(T_K^e, T^*)$ (although $T_{FL}$ itself may have a nontrivial gate dependence). At large $\delta \sim J_+$ ($T^* \gg T_K^e$), the behaviour of the single-channel AIM[20,52] is essentially recovered for the entire crossover (see red line, Fig. 3b). However, the universality of the AIM is lost for smaller $\delta$ due

to the competing involvement of the odd screening channel. In fact, for $T^* \ll T_K^e$, appreciable conductance only sets in around $T \sim T^*$ (rather than $T_K^e$), and the entire conductance crossover becomes a universal function of $T/T^*$—different in form from that of the AIM (see black line, Fig. 3b). The formation of the Kondo state is reflected in conductance by the following limiting behaviour,

$$G(T) \overset{T \gg T_{FL}}{\sim} \begin{cases} \ln^{-2}|T/T_K^e| & : \quad T^* \gg T_K^e, \\ (T/T^*)^{-1} & : \quad T^* \ll T_K^e. \end{cases} \quad (6)$$

Furthermore, the abelian bosonization methods of refs 53–55 can be applied to single-molecule junctions in the limit $T^* \ll T_K^e$ to obtain an exact analytic expression for the full conductance crossover (Supplementary Note 3),

$$G(T, V_{sd})/G_0 = \frac{T^*}{2\pi T} \operatorname{Re} \psi_1\left(\frac{1}{2} + \frac{T^*}{2\pi T} + i\frac{eV_{sd}}{2\pi k_B T}\right), \quad (7)$$

where $\psi_1$ is the trigamma function. Remarkably, this result also holds away from thermal equilibrium, at finite bias $V_{sd} \ll T_K^e$. Within linear response, equation (7) is confirmed explicitly by comparison to NRG data in Fig. 3b, while Fig. 3c shows the nonequilibrium predictions. The condition $T^* \ll T_K^e$ pertains to nearly symmetric junctions, tuned near a QI node in $J_{sd}$. Equation (7) should be regarded as a limiting scenario: conductance lineshapes for real single-molecule junctions will typically interpolate between the red and black lines of Fig. 3b.

**Kondo blockade**. At a QI node in the exchange-cotunneling $J_{sd} = 0$, conductance through a single-molecule junction is mediated solely by $W_{sd}$. In this case, the molecule spin is fully Kondo screened by either the source or drain lead (whichever is more strongly coupled). Only in the special but unrealistic case $J_{ss} = J_{dd}$ and $J_{sd} = 0$ does the frustration persist down to $T = 0$. For concreteness we now assume ph symmetry $W_{ss} = W_{dd} = 0$, and $J_{ss} > J_{dd}$ such that the even conduction electron channel is simply the source lead. The drain lead therefore decouples on the scale of $T_K^s$. As shown in Supplementary Note 4, one can then prove that,

$$G(T=0) = G_0 (2\pi\rho_0 W_{sd})^2 [1 - t_{ss}(0,0)] = 0, \quad (8)$$

where $t_{ss}(\omega, T)$ is the T-matrix of the source lead. The Kondo effect with the source lead, characterized by $t_{ss}(0,0) = 1$, therefore exactly blocks current flowing from source to drain. This is an emergent effect of interactions—at high temperatures $T \gg T_K$

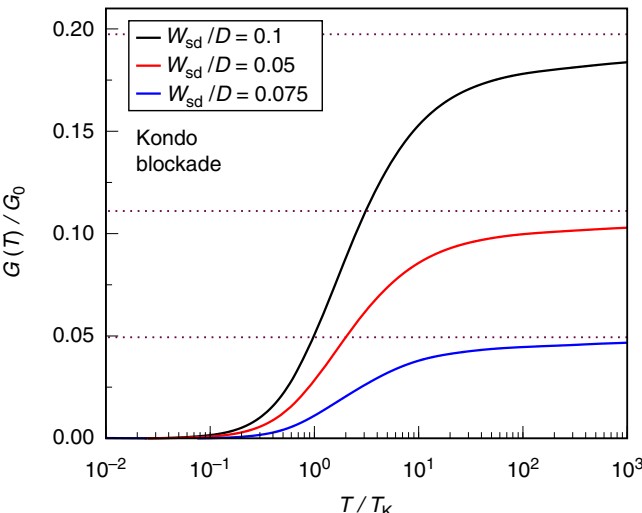

**Figure 4 | Kondo blockaded conductance near a quantum interference node in the exchange cotunneling.** NRG results for the conductance $G(T)$ as a function of rescaled temperature $T/T_K$ at $J_{sd} = 0$ for various $W_{sd}$, showing in all cases an overall conductance node $G(0) = 0$. Plotted for $J_{ss} = 0.25D$, $J_{dd} = 0.2D$ and $W_{ss} = W_{dd} = 0$. Dotted lines show the high-temperature perturbative expectation.

when $t_{ss} \approx 0$, conductance is finite and the perturbative result is recovered, $G_{pert}/G_0 \approx (2\pi\rho_0 W_{sd})^2$.

The zero-temperature conductance node arising for $J_{sd} = 0$ can be understood physically, as a depletion of the local source-lead density of states at the junction, due to the Kondo effect. Conductance vanishes because locally, no source-lead states are available from which electrons can tunnel into the drain lead. From a real-space perspective[17], one can think of the Kondo cloud in the source lead as being impenetrable to electronic tunnelling at low energies. The effect of this Kondo blockade is demonstrated in Fig. 4, where full NRG calculations for the conductance are shown for $J_{sd} = 0$. The conductance crossover as a function of temperature is entirely characteristic of the Kondo effect; $G(T)/G_{pert}$ is a universal function of $T/T_K$. At low temperatures, a node in $J_{sd}$ thus implies an overall conductance node, even though $W_{sd}$ remains finite.

The Kondo blockade will be most cleanly observed in real single-molecule junctions that have strong molecule-lead hybridization and do not have a nearby Kondo resonance. In addition to the large Kondo temperature, the perturbative cotunneling conductance observed at high temperatures $T \gg T_K$ is also larger in this case, thereby increasing the contrast of the blockade on lowering the temperature.

We emphasize that the Kondo blockade is unrelated to the Fano effect[34,56,57], which arises due to QI in the hybridization rather than intrinsic QI in the interacting molecule itself (Supplementary Note 4). Unlike the Kondo blockade, the Fano effect is essentially a single-channel phenomenon that does not necessitate interactions, and different (asymmetric) lineshapes result.

**Gate-tunable QI in Kondo-active molecules.** In real molecular junctions, the two conductance mechanisms discussed separately above (due to finite exchange $J_{sd}$ and potential scattering $W_{sd}$), are typically both operative. Their mutual effect can be complicated due to renormalization from cross-terms proportional to $W_{sd} J_{sd}$. However, as the gate voltage $V_g$ is tuned, both Kondo resonant and Kondo blockade regimes are often accessible due to QI nodes[45] in either $J_{sd}$ or $W_{sd}$. In practice, we observe that

overall conductance nodes can also be shifted away from the nodes in $J_{sd}$ by marginal potential scattering $W_{ss}$ and $W_{dd}$ (not considered above). We speculate that the conductance nodes are topological and cannot be removed by potential scattering—only shifted to a different gate voltage. Precisely at the node, the low-temperature physics is universal and therefore common to all such off-resonant spin-$\frac{1}{2}$ molecules.

To demonstrate the gate-tunable interplay between QI and the Kondo effect in single-molecule junctions, we now consider two simple conjugated organic molecules as examples. Following ref. 45, exact diagonalization of the Pariser–Parr–Pople (PPP) model[58] for the $sp^2$-hybridized $\pi$ system of the molecule allows the effective 2CK model parameters to be extracted as a function of applied gate voltage (Supplementary Note 5). The 2CK model is then solved using NRG[26,27], and the conductance is calculated numerically-exactly as a function of temperature. These steps are described in detail in Methods.

Figure 5 shows the conductance $G(T)$ for junctions spanned by respectively a benzyl, (a) and an isoprene-like molecule (d), as a function of rescaled temperature $T/T_K$ at different gate voltages. Both systems exhibit Kondo resonant and Kondo blockade physics. In panel (a), a pronounced Kondo blockade appears near $V_g = 0$, corresponding to the midpoint of the Coulomb diamond. Finite conductance at higher temperatures due to cotunneling $W_{sd}$ is blocked at low temperatures by the Kondo effect. On increasing the gate voltage, we find numerically that $G(0) \sim eV_g^2$, with conductance enhancement due to renormalized $J_{sd}$ (Fig. 5b). The overall conductance in this case remains rather small for all $eV_g$ analysed. We also note that the Kondo temperature varies as $\ln T_K/D \sim eV_g^2$, Fig. 5c. This gate evolution of $T_K$ could be considered as conventional from the single-orbital AIM perspective[10], but the conductance itself is blockaded rather than enhanced by Kondo correlations.

However, richer physics can be accessed in junction (d). The crossovers of $G(T)$ show perfect Kondo resonant conductance at finite $eV_g = 2.4$ eV, reaching the unitarity limit $G(0) = 2e^2h^{-1}$. But increasing the gate voltage slightly to $V_g = 2.625$ eV yields almost perfect Kondo blockade, with $G(0) \simeq 0$ (note the log scale). The full crossovers are entirely characteristic of the underlying correlated electron physics. Panel (e) shows the evolution of $G(0)$ as a function of gate voltage at $T = 0$ (and in practice for all $T \ll T_K$), which exhibits nontrivial behaviour due to the interplay between QI and the Kondo effect. The rapid switching between Kondo resonant and Kondo blockade conductance with applied gate voltage might make such systems candidates for QI-effect transistors, or other technological applications.

Finally, in panel (f), we show that the Kondo temperature also displays an unconventional gate-dependence, with $T_K$ increasing as one moves in towards $eV_g = 0$, analogous to the effect observed experimentally in ref. 28. The Kondo temperature remains finite for all $eV_g$, but takes its minimum value at the Kondo resonance peak. In practice, the Kondo temperature can vary widely from system to system because it depends sensitively (exponentially) on the molecule-lead hybridization. However, Kondo temperatures up to around 30 K are commonly observed in real single-molecule junctions[10].

We did not attempt an *ab initio* calculation of the absolute Kondo temperatures, but note that the effective bandwidth cutoff $D$ in the effective 2CK model is essentially set by the large charging energy of the molecule. For the PPP models used for the conjugated hydrocarbons in Fig. 5, this in turn is set by the onsite Coulomb repulsion, taken to be 11 eV within the standard Ohno parametrization[58]. With this identification, we have $T_K \sim 10$ K for the specific example shown in panel (a) at the Kondo blockade, and 0.1 K in (d). We emphasize that the Kondo blockade arises on similar temperature scales to that of the standard Kondo effect in

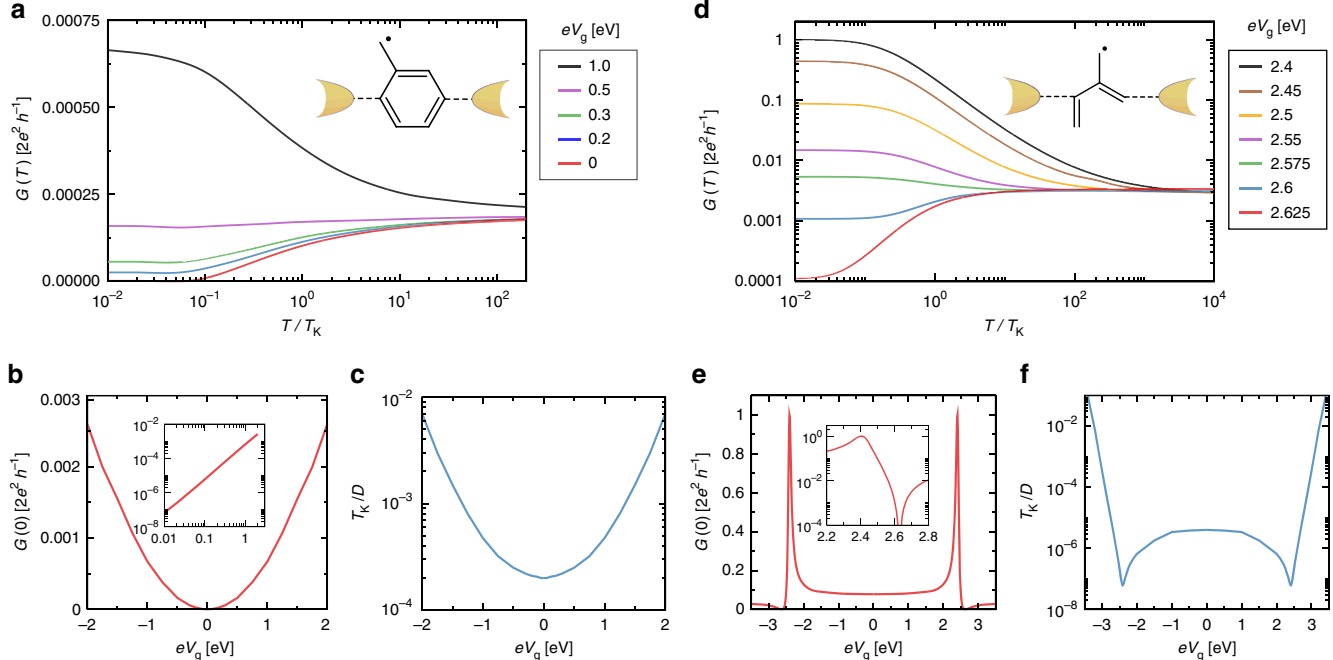

**Figure 5 | Gate-tunable Kondo resonance and Kondo blockade in simple conjugated organic molecular junctions.** Single-molecule junctions based on a benzyl (**a–c**) and an isoprene-like molecule (**d–f**), with all carbons $sp^2$-hybridized, were mapped to an effective 2CK model and linear conductance was calculated with NRG. (**a,d**) Conductance $G(T)$ as a function of rescaled temperature $T/T_K$ for various gate voltages $eV_g$. (**a**) shows Kondo blockade $G(0) = 0$ at $eV_g = 0$ and Kondo enhanced conductance for $|V_g| > 0$. (**d**) shows Kondo blockade at $eV_g = 2.625$ eV and perfect (unitarity) Kondo resonance at $eV_g = 2.4$ eV. (**b,e**) $G(0)$ as a function of gate voltage $eV_g$ at $T = 0$; (**c,f**) Corresponding Kondo temperatures. Note the sensitive gate dependence of $G(0)$ in (**e**), and the corresponding unconventional non-monotonic gate dependence of the Kondo temperature in (**f**).

molecules, and therefore signatures should generally be observable at experimental base temperatures on gate tuning to a QI node.

The stark difference in transport properties of the two molecular junctions shown in Fig. 5 is due to differences in their QI characteristics—specifically the number and position of QI nodes in the effective 2CK parameters (Supplementary Note 5). In turn, this is related to the underlying molecular structure and contacting geometry, as explored for these alternant hydrocarbons in ref. 45. Both molecules exhibit a Kondo blockade due to a node in $J_{sd}$, but this arises at $eV_g = 0$ for the benzyl radical in (**a–c**), whereas there are two nodes at finite $\pm eV_g$ for the isoprene-like molecule in (**d–f**). In general, $J_{sd}$ has an odd(even) number of nodes as a function of gate in odd-membered alternant molecules if the source and drain electrodes are connected to sites of the molecule on different sublattices(the same sublattice) of the bipartite $\pi$ system. The strong Kondo resonance arising at $eV_g = 2.4$ eV for the isoprene-like molecule is a consequence of the parity symmetry with respect to the contacting geometry, such that $J_{ss} \approx J_{dd}$ (equation (5)). By contrast, there is no such symmetry for the benzyl molecule and $J_{dd}$ happens to dominate.

Although we have exemplified the gate-tunable interplay between QI and Kondo effect with these conjugated hydrocarbon moieties, we emphasize that a Kondo blockade should be found in any off-resonant spin-$\frac{1}{2}$ molecule with intramolecular interactions and sufficient orbital complexity to produce a QI node in the exchange cotunneling.

## Discussion

Transport through spinful Coulomb-blockaded single-molecule junctions requires a description beyond the standard single-orbital Anderson paradigm. The relevant model is instead a generalized two-channel Kondo model, to which real molecular junctions can be exactly mapped. Experimental data for individual molecular junctions can be understood within this framework, avoiding the need for a statistical interpretation.

Quantum interference can be classified as being of either exchange or potential scattering type. Although these distinct conductance mechanisms are simply additive at high energies, where standard perturbation theory holds, the low-temperature behaviour is much richer due to electron–electron interactions which drive the Kondo effect. We show that the Kondo effect survives a quantum interference node in the potential scattering to give enhanced conductance, while a novel Kondo blockade arises in the case of an exchange cotunneling node, entirely blocking the current through the junction. This rich physics is tunable by applying a back-gate voltage, as demonstrated explicitly for two simple conjugated organic molecules, opening up the possibility of efficient Kondo-mediated quantum interference effect transistors.

The theoretical framework we present can be used to systematically study candidate molecules and help optimize the type and location of anchor groups for particular applications. Quantum chemistry techniques could be used to accomplish the Kondo model mapping for larger molecules. The effect of vibrations and dissipation (relevant at higher energies[5]) could also be taken into account within generalized Anderson–Holstein or Bose–Fermi Kondo models[27].

## Methods

**Schrieffer-Wolff transformation.** We derive the effective Kondo model describing off-resonant single-molecule junctions by projecting out high-energy molecular charge fluctuations from the full lead-coupled system. This is equivalent to a two-channel generalization of the standard Schrieffer–Wolff transformation[19]. That is, projecting onto the subspace of Hilbert space where the number, $N$, of electrons on the molecule is fixed. The effective Hamiltonian in this subspace has

the form

$$H_{eff}(E) = P\big[H_{leads} + H_{mol} + H_g + H_{hyb}Q(E - QHQ)^{-1}QH_{hyb}\big]P, \qquad (9)$$

where $P$ is a projection operator onto the $N$-electron subspace of the molecule, while $Q = I - P$ projects onto the orthogonal complement. These subspaces are connected by $H_{hyb}$, and the resolvent operator $(E - QHQ)^{-1}$ determines the propagation of excited states at energy $E$. Note that $PH_{leads} = H_{leads}$ since $P$ acts only on the molecular degrees of freedom, and $P(H_{mol} + H_g)P$ is merely a constant and dropped in the following. For the isolated molecule, $H_{mol}|\Psi_n^N\rangle = E_n^N|\Psi_n^N\rangle$, where $|\Psi_n^N\rangle$ denotes the $n$'th $N$-electron many-body eigenstate with energy $E_n^N$, and $E_0^N$ is the ground state energy. $|\Psi_n^N\rangle$ spans the entire molecule and generally has weight on all atomic/molecular basis orbitals in $H_{mol}$. So far the treatment is exact.

To second order in $H_{hyb}$, Equation (9) reduces to the effective Hamiltonian

$$H_{eff} = H_{leads} + PH_{hyb}Q\big(E_0^N - Q(H_{mol} + H_g)Q\big)^{-1}QH_{hyb}P. \qquad (10)$$

Virtual processes involving a given excited state $|\Psi_n^{N\pm1}\rangle$ contribute to Equation (10) with weight controlled by the energy denominator $\langle\Psi_n^{N\pm1}|E_0^N - Q(H_{mol} + H_g)Q|\Psi_n^{N\pm1}\rangle = E_0^N - E_n^{N\pm1} \pm eV_g$, which must be negative to ensure stability of the $N$-electron ground state (including the electrostatic shift from the backgate described by $H_g$). The perturbative expansion in $H_{hyb}$ is controlled by a large energy denominator, and therefore use of Equation (10) is justified deep inside the $N$-electron Coulomb diamond. Inserting the tunnelling Hamiltonian, $H_{hyb} = \sum_{\alpha\sigma k}(t_{\alpha k}d_{i_\alpha\sigma}^\dagger c_{\alpha k\sigma} + \text{H.c.})$, one arrives at the effective Hamiltonian $H_{eff} = H_{leads} + H_{ex}$, with

$$H_{ex} = \sum_{\substack{\alpha'k'\sigma' \\ \alpha k\sigma}} t_{\alpha'k'}^* t_{\alpha k}^* c_{\alpha k\sigma}^\dagger c_{\alpha'k'\sigma'} \sum_{m',m}|\Psi_{m'}^N\rangle A_{\sigma'\sigma,m'm}^{\alpha'\alpha}\langle\Psi_m^N|$$
$$= \sum_{\substack{\alpha'\sigma' \\ \alpha\sigma}} t_{\alpha'}^* t_\alpha^* c_{\alpha\sigma}^\dagger c_{\alpha'\sigma'} \sum_{m',m}|\Psi_{m'}^N\rangle A_{\sigma'\sigma,m'm}^{\alpha'\alpha}\langle\Psi_m^N|, \qquad (11)$$

where the last line follows from the definition of local lead-electron operators $c_{\alpha\sigma} = t_\alpha^{-1}\sum_k t_{\alpha k}c_{\alpha k\sigma}$, with $t_\alpha^2 = \sum_k|t_{\alpha k}|^2$. Here $m$ and $m'$ label (degenerate) molecular ground states with energy $E_0^N$. For odd $N$, the molecule often carries a net spin-$\frac{1}{2}$, and so $m$ and $m'$ are simply the projections $S^z = \pm\frac{1}{2}$. The spin density need not be spatially localized. We now focus on this standard case, although the generalization to arbitrary spin is straightforward when the molecular ground state for a given $N$ is more than two-fold degenerate.

The cotunneling amplitudes can be decomposed as,

$$A_{\sigma'\sigma,m'm}^{\alpha'\alpha} = h_{\sigma'\sigma,m'm}^{\alpha'\alpha} + p_{\sigma'\sigma,m'm}^{\alpha'\alpha} \qquad (12)$$

where the contributions from hole and particle propagation are given respectively by,

$$h_{\sigma'\sigma,m'm}^{\alpha'\alpha}(V_g) = \sum_n \frac{\langle\Psi_{m'}^N|d_{i_{\alpha'}\sigma'}^\dagger|\Psi_n^{N-1}\rangle\langle\Psi_n^{N-1}|d_{i_\alpha\sigma}|\Psi_m^N\rangle}{eV_g - E_0^N + E_n^{N-1} - i0^+}, \qquad (13)$$

$$p_{\sigma'\sigma,m'm}^{\alpha'\alpha}(V_g) = \sum_n \frac{\langle\Psi_{m'}^N|d_{i_\alpha\sigma}|\Psi_n^{N+1}\rangle\langle\Psi_n^{N+1}|d_{i_{\alpha'}\sigma'}^\dagger|\Psi_m^N\rangle}{eV_g + E_0^N - E_n^{N+1} + i0^+}. \qquad (14)$$

The matrix elements in the numerators (referred to as Feynman-Dyson orbitals) constitute a correlated generalization of molecular orbitals, and are computed in the many-particle molecular eigenstate basis.

Since the total Hamiltonian must preserve its original spin-rotational invariance, the cotunneling amplitude must take the form

$$A_{\sigma'\sigma,m'm}^{\alpha'\alpha} = \tfrac{1}{4}J_{\alpha'\alpha}\,\boldsymbol{\tau}_{\sigma'\sigma}\cdot\boldsymbol{\tau}_{mm'} + W_{\alpha'\alpha}\delta_{\sigma'\sigma}\delta_{mm'}. \qquad (15)$$

This leads to the desired effective 2CK model (equation (2) of subsection 'Models and mappings'):

$$H_{2CK} = H_{leads} + \sum_{\substack{\alpha'\alpha \\ \alpha\sigma}}\big(\tfrac{1}{2}J_{\alpha'\alpha}\mathbf{S}\cdot\boldsymbol{\tau}_{\sigma'\sigma} + W_{\alpha'\alpha}\delta_{\sigma'\sigma}\big)c_{\alpha'\sigma'}^\dagger c_{\alpha\sigma}. \qquad (16)$$

The 2CK model parameters themselves are obtained from traces with Pauli matrices

$$J_{\alpha\alpha'} = t_{\alpha'}^* t_\alpha^* \sum_{\sigma\sigma',mm'}\tau_{\sigma'\sigma}^i A_{\sigma'\sigma,m'm}^{\alpha'\alpha}\tau_{m'm}^i \text{ for } i = x, y, z, \qquad (17)$$

$$W_{\alpha\alpha'} = t_{\alpha'}^* t_\alpha^* \sum_{\sigma\sigma',mm'} A_{\sigma\sigma,mm}^{\alpha'\alpha} \qquad (18)$$

which, by spin-rotation invariance, further simplify to

$$J_{\alpha\alpha'} = 2t_{\alpha'}^* t_\alpha^* A_{\uparrow\downarrow,\downarrow\uparrow}^{\alpha'\alpha} = 2t_{\alpha'}^* t_\alpha^* \sum_m A_{\uparrow\uparrow,mm}^{\alpha'\alpha}\tau_{mm}^z \qquad (19)$$

$$W_{\alpha\alpha'} = 4t_{\alpha'}^* t_\alpha^* A_{\uparrow\uparrow,\uparrow\uparrow}^{\alpha'\alpha} - 2A_{\uparrow\downarrow,\downarrow\uparrow}^{\alpha'\alpha} = 2t_{\alpha'}^* t_\alpha^* \sum_m A_{\uparrow\uparrow,mm}^{\alpha'\alpha} \qquad (20)$$

such that in practice only two matrix elements are needed to obtain the exchange couplings $J_{\alpha\alpha'}$ and the potential scattering amplitudes $W_{\alpha\alpha'}$.

Equations (13) and (14) therefore encode all the properties of the single-molecule junction inside an $N$-electron Coulomb diamond. The exchange and potential scattering terms in the 2CK model are determined by the amplitudes $A$ which, from equation (12), have contributions from both particle ($p$) and hole ($h$) processes (that is, processes involving virtual states with $N+1$ or $N-1$ electrons on the molecule). In particular, note that all quantum interference effects are entirely encoded in the effective parameters $J_{\alpha\alpha'}$ and $W_{\alpha\alpha'}$—quantum interference nodes arise if and only if molecular states are connected by particle and hole processes with equal but opposite amplitudes. As discussed in ref. 45, the appearance of such nodes can be understood in terms of the properties of the underlying Feynman–Dyson orbitals.

We emphasize that the effective 2CK model is totally general, applying for any molecule with a two-fold spin-degenerate ground state, at temperatures less than the molecule charging energy so that charge fluctuations on the molecule are frozen (typically the charging energy is large when deep inside a Coulomb diamond, and therefore the molecule is off-resonant). The 2CK model parameters can be obtained purely from a knowledge of the isolated molecule, and can therefore be calculated in practice using a number of established techniques (exact diagonalization, configuration interaction and so on). The low-temperature properties of the resulting 2CK model are however deeply nontrivial, requiring sophisticated many-body methods to 'attach the leads' and account for nonperturbative renormalization effects. In the present work, we do this second step using the numerical renormalization group[27].

An advantage of the effective theory is that it can also be analysed exactly on an abstract level (independently of any specific realization). This allows us to identify all the possible scenarios that could in principle arise in molecular junctions. The basic physics is arguably obfuscated rather than clarified by the complexity of a full microscopic description: a brute-force method (even if that were possible) may not yield new conceptual understanding or provide general predictions beyond a case-by-case basis.

2CK parameters for the molecules presented in Fig. 5 were obtained following ref. 45; see Supplementary Figs 1 and 2.

**Exact diagonalization of $H_{mol}$.** In this work, we model the isolated molecule by a semi-empirical Pariser-Parr-Pople Hamiltonian[59,60] for the molecular $\pi$-system:

$$\hat{H}_{mol} = \sum_{\langle i,j\rangle}\sum_{\sigma=\uparrow/\downarrow}\big(t_{ij}d_{i\sigma}^\dagger d_{j\sigma} + \text{H.c.}\big) + \sum_i U\big(n_{i\uparrow} - \tfrac{1}{2}\big)\big(n_{i\downarrow} - \tfrac{1}{2}\big) + \tfrac{1}{2}\sum_{i\neq j}V_{ij}(n_i - 1)(n_j - 1). \qquad (21)$$

The operator $d_{i\sigma}^\dagger$ creates an electron with spin $\sigma$ on the $p_z$-orbital $|i\rangle$, $n_{i\sigma} = d_{i\sigma}^\dagger d_{i\sigma}$ and $n_i = n_{i\uparrow} + n_{i\downarrow}$. The Coulomb interaction is given by the Ohno parametrization[58] $V_{ij} = U/(\sqrt{1 + |\vec{r}_{ij}|^2 U^2/207.3\,\text{eV}})$, where $|\vec{r}_{ij}|$ is the real-space distance between two $p_z$-orbitals $|i\rangle$ and $|j\rangle$ measured in Ångström. For $sp^2$ hybridized carbon, the nearest neighbour overlap, $t_{ij}$, is $t \approx -2.4\,\text{eV}$, and $U \approx 11.26\,\text{eV}$ (ref. 61).

For suitably small molecules, equation (21) can be solved using exact diagonalization (exploiting overall conserved charge and spin) to provide the many-particle eigenstates $|\Psi_n^N\rangle$ and eigenenergies $E_n^N$. For larger molecules, approximate methods can also be used, provided interactions are accounted for on some level. Any molecule can be addressed within our framework, provided the eigenstates and eigenenergies of the isolated molecule can be determined.

Importantly, the perturbed two-channel Kondo model derived in the previous section remains the generic Hamiltonian of interest to describe such off-resonant junctions. The diagonalization of equation (21) is required only to obtain the parameters $J$ and $W$, which are then used in subsequent numerical renormalization group calculations to treat the coupling to source and drain leads. Note that the calculation of physical quantities such as conductance at lower temperatures necessitates an explicit and nonperturbative treatment of the leads, and cannot be achieved with single-particle methods or exact diagonalization alone.

However, once the generic physics of the underlying 2CK model is understood (a key goal of this paper), the transport properties and quantum interference effects of specific molecular junctions can already be rationalized and predicted from their 2CK parameters. The suitability of candidate molecules and the positions of anchor groups can therefore be efficiently assessed, opening up the possibility of rational device design.

**Calculation of conductance.** The key experimental quantity of interest for single-molecule junction devices is the differential conductance $G(T, V_{sd}) = d\langle I_{sd}\rangle/dV_{sd}$. In this section we recap the generic framework for exact calculations of the linear response conductance $G(T) \equiv G(T, V_{sd} \to 0)$ through a molecule, taking fully into account renormalization effects due to electronic interactions. We then describe how the NRG[27] can be used to accurately obtain $G(T)$ for a given system described by the effective model, equation (16).

To simulate the experimental protocol, we add a time-dependent bias term to the Hamiltonian, $H = H_{2CK} + H'(t)$, with

$$H'(t) = \frac{eV_{sd}}{2}\cos(\omega t)\big(\hat{N}_s - \hat{N}_d\big), \qquad (22)$$

where $\hat{N}_\alpha = \sum_{k,\sigma}c_{\alpha k\sigma}^\dagger c_{\alpha k\sigma}$ is the total number operator for lead $\alpha$. We focus on

the serial ac and dc conductance at $t = 0$, after the system has reached an oscillating steady state. In the zero-bias limit $V_{sd} \to 0$, the exact serial ac conductance at linear response follows from the Kubo formula[62],

$$G^{ac}(\omega, T) = \frac{e^2}{h} \times \left[ \frac{-2\pi\hbar^2 \operatorname{Im} K(\omega, T)}{\hbar\omega} \right], \qquad (23)$$

where $K(\omega, T)$ is the Fourier transform of the retarded current–current correlator,

$$K(t, T) = i\theta(t) \left\langle \left[ \hat{\Omega}(t), \hat{\Omega}(0) \right] \right\rangle_T, \qquad (24)$$

where $\hat{\Omega} = \frac{1}{2}(\dot{N}_s - \dot{N}_d)$ and $\dot{N}_\alpha = \frac{d}{dt}\hat{N}_\alpha$. The dc conductance is then simply,

$$G(T) = \lim_{\omega \to 0} G^{ac}(\omega, T). \qquad (25)$$

In practice, NRG is used to obtain $K(\omega, T)$ numerically. The full density matrix NRG method[63], established on the complete Anders–Schiller basis[64], provides essentially exact access to such dynamical correlation functions at any temperature $T$ and energy scale $\omega$. We use $\dot{N}_\alpha = i[\hat{H}, \hat{N}_\alpha]$ to find an expression for the current operator amenable for treatment with NRG:

$$i\hat{\Omega} = \left( J_{sd}\mathbf{S} \cdot \mathbf{s}_{sd} + W_{sd} \sum_\sigma c^\dagger_{s\sigma} c_{d\sigma} \right) - \text{H.c.}, \qquad (26)$$

where $\mathbf{s}_{\alpha\beta} = \frac{1}{2} \sum_{\sigma\sigma'} c^\dagger_{\alpha\sigma} \boldsymbol{\sigma}_{\sigma\sigma'} c_{\beta\sigma'}$.

As the ground state of any conducting molecular junction must be a Fermi liquid, the system can be viewed as a renormalized non-interacting system at $T = 0$ (ref. 65). The zero-bias dc conductance at $T = 0$ can therefore also be obtained from a Landauer–Büttiker treatment,

$$G(T=0) = \frac{2e^2}{h} \times 4\tilde{\Gamma}_s \tilde{\Gamma}_d |\mathcal{G}_{sd}(\omega = 0, T = 0)|^2, \qquad (27)$$

in terms of the full retarded electronic Green's function $\mathcal{G}_{\alpha\beta}(\omega, T) \overset{\text{FT}}{\leftrightarrow} -i\theta(t) \langle \{ c_{\alpha\sigma}(t), c^\dagger_{\beta\sigma}(0) \} \rangle_T$, which must be calculated non-perturbatively in the presence of the interacting molecule. Here $\tilde{\Gamma}_\alpha = 1/(\pi\rho_0)$. Note that equation (27) applies to Fermi liquid systems only at $T = 0$ and in the dc limit. The full temperature dependence of $G(T)$ must be obtained from the Kubo formula.

In practice, $\mathcal{G}_{sd}(\omega, T)$ is obtained from the T-matrix equation[19], which describes electronic scattering in the leads due to the molecule,

$$\mathcal{G}_{\alpha\beta}(\omega, T) = \mathcal{G}^{(0)}(\omega)\delta_{\alpha\beta} + \left[ \mathcal{G}^{(0)}(\omega) \right]^2 \times \left[ W_{\alpha\beta} + T_{\alpha\beta}(\omega, T) \right], \qquad (28)$$

where $\mathcal{G}^{(0)}(\omega)$ is the free retarded lead electron Green's function when the molecule is disconnected, such that $\operatorname{Im} \mathcal{G}^{(0)}(\omega) = -\pi\rho(\omega)$, and $\rho(0) = \rho_0$.

Within NRG, the T-matrix can be calculated directly[51] as the retarded correlator $T_{\alpha\beta}(\omega, T) \overset{\text{FT}}{\leftrightarrow} -i\theta(t)\langle\{a_\alpha(t), a^\dagger_\beta(0)\}\rangle_T$. For the present problem, the composite operators,

$$a_\alpha = \sum_\gamma \left[ W_{\alpha\gamma} c_{\gamma\uparrow} + \frac{1}{2} J_{\alpha\gamma} \left( c_{\gamma\uparrow} S^z + c_{\gamma\downarrow} S^- \right) \right], \qquad (29)$$

follow from equation (16) using equations of motion methods. In subsection 'Emergent decoupling' we also present NRG results for the spectrum of the T-matrix, defined as

$$t_{\alpha\beta}(\omega, T) = -\pi\rho_0 \operatorname{Im} T_{\alpha\beta}(\omega, T). \qquad (30)$$

Note that the simple Landauer form of the Meir–Wingreen formula[66], which relates the conductance through an interacting region to a generalized transmission function, applies only in the special case of proportionate couplings. In single-molecule junctions, the various molecular degrees of freedom couple differently to source and drain leads (which are spatially separated), and therefore this standard form of the Meir–Wingreen formula cannot be used, and one has to resort to using full Keldysh Green's functions (or the methods described above for linear response). The exception is when the molecule is a single orbital—this artificial limit is considered in Supplementary Note 1.

For the NRG calculations, even/odd conduction electron baths were discretized logarithmically using $\Lambda = 2$, and $N_s = 15,000$ states were retained at each step of the iterative procedure. Total charge and spin projection quantum numbers were exploited to block-diagonalize the NRG Hamiltonians, and the results of $N_z = 2$ calculations were averaged. 2CK model parameters for the molecules presented in Fig. 5 are discussed in Supplementary Note 5.

**Data availability.** The data supporting our findings are available from the corresponding author on reasonable request.

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

## Acknowledgements

We thank Eran Sela and Martin Galpin for fruitful discussions. A.K.M. acknowledges funding from the D-ITP consortium, a program of the Netherlands Organisation for Scientific Research (NWO) that is funded by the Dutch Ministry of Education, Culture and Science (OCW). The Center for Quantum Devices is funded by the Danish National Research Foundation. We are grateful for use of HPC resources at the University of Cologne.

## Author contributions

A.K.M. wrote the NRG code, performed NRG calculations and derived analytic results. K.G.L.P., P.H. and J.P. formulated the 2CK mapping. K.G.L.P derived effective model parameters. All authors prepared the manuscript.

## Additional information

**Competing interests:** The authors declare no competing financial interests.

**Publisher's note**: 

