## [Peer Review File · Nature Communications]

Reviewers' Comments:

Reviewer #1 (Remarks to the Author)

The paper “Kondo blockade due to quantum interference in single-molecule junctions” is devoted to low-temperature transport properties of molecules exhibiting strong correlations that can give rise to the Kondo effect. In particular, the Authors argue that a novel Kondo blockade effect associated with quantum interference can emerge in such systems. The Authors have an extensive experience on analyzing transport through Kondo-correlated junctions using both analytical and numerical (such as NRG) methods, and in this paper they present a sound and serious study performed for real molecules. The single-molecule is modeled by the two-channel Kondo model (2CK), the derivation of which is presented in supplementary material. Although the physics of 2CK is relatively well understood, the new aspects of this work are related with shedding more light on the interplay and competition between quantum interference and Kondo correlations in real molecular junctions. Such interplay may give rise to full conductance suppression, which the Authors named “Kondo blockade”. In this blockade, the Kondo singlet develops between the molecule and one of the leads, prohibiting the electrons from tunneling between the source and drain leads.

I find the paper clearly written and well-structured. However, before proceeding with publication, I'd like the Authors to consider the following points:

1. I believe the whole study is devoted to spin $-1/2$ molecules. This is explicitly stated in “the molecule typically hosts a spin- $1/2$ degree of freedom”. However, from this statement one may infer that the analysis is also valid for larger-spin molecules, which is not necessarily true. In particular, for systems with $S > 1/2$, one could expect underscreened or fully-screened Kondo effects to emerge. The physics then may be completely different. This should be emphasized in the paper.
2. I think it may be worthwhile to introduce the notion of chain-like configuration (Wilson chain), when discussing the illustrations and Kondo screening clouds in Fig.2. Since from the Hamiltonian, Eq. (1), one cannot guess the chain configuration shown in Fig. 2, this will improve readability.
3. The “Kondo blockade” results from the suppression of T-matrix when the coupling to one of the channels wins. The full energy-dependence of the T-matrix for 2CK was calculated in Phys. Rev. B 76, 155318 (2007). Also the frequency-dependent conductance, Eq. (29) in Suppl.

Material, was derived in this paper. It would be good to add a reference to this paper.

4. In Fig. 5(c) and (e): at what T was the conductance calculated? There is a typo in the caption: Kondo blockade develops for $V_g = 2.625\text{eV}$ and the resonance for $V_g = 2.4\text{eV}$. Please make sure that the units for V_g are all the same: in axis label it is in V, in legend to (a) and (b) it is in eV.

5. It may be good to estimate T_K in Kelvins in Fig. 5: What is the temperature range to observe the Kondo blockade ($T < 0.1T_K$), is it experimentally accessible?

6. I do not understand the conductance enhancement in Fig. 5(e): is it due to the Kondo effect? I guess so, since this is the conductance for Coulomb blockade of spin $1/2$ system. However, at V_g where $G = 2e^2/h$, $T_K=0$, such that there is no Kondo effect. Could the Authors explain this behavior in more detail?

7. The conductance suppression due to the Kondo blockade (Fig. 5) looks very small, I wonder if it would be experimentally measurable at all. Can it be enhanced in other systems? Could the Authors comment on that?

8. When referring to NRG method, I encourage the Authors to cite also the original Wilson's paper.

9. A similar conductance suppression due to quantum interference was also observed in double quantum dot systems, see e.g. Phys. Rev. Lett. 87, 216601(2001), Phys. Rev. Lett. 103, 266806 (2009), Phys. Rev. B 81, 115316 (2010). The conductance suppression can be understood there by invoking the two-stage Kondo effect and is therefore also an example of strong-correlation-mediated current blockade.

Reviewer #2 (Remarks to the Author)

The article "Kondo blockade due to quantum interference in single-molecule junctions" by Andrew K. Mitchel et al. presents a theoretical study of the interplay between quantum interference and Kondo effect in single-molecule junctions carrying a spin-1/2, and predict as a result of this interplay a novel phenomenon to occur in these kind of junctions, the Kondo blockade. Most of the results presented in the paper are for a simplified effective model of a molecular junction, i. e. the two-channel spin-1/2 Kondo (2CK) model where the spin-1/2 represents an effective model of the molecule, while the two conduction electron channels exchange coupled to the spin-1/2, represent the metallic (source and drain) leads connected to the molecule. In the second part of the paper and in the Supplementary Material the authors show, using the well-known Schrieffer-Wolff transformation how this effective model emerges from more realistic models of a molecular junctions when the junction is off resonance, corresponding

to the limit of frozen charge fluctuations. The results for the 2CK model are obtained by Numerical Renormalization Group (NRG) calculations, which are numerically exact and considered the gold standard for the solution of Anderson and Kondo impurity models.

The paper is generally well written and the presented results and predictions are interesting and possibly relevant for the understanding of real molecular junction experiments. However, there are a few things that are not entirely clear to me and that the authors should clarify:

1) The "Kondo blockade" occurs at an interference node in the source-drain exchange coupling ($J_{sd}=0$). As far as I understand, in this limit the model should be governed by two-channel Kondo physics down to zero temperature, leading to frustration due to the competition between even and odd screening channels, as explained by one of the authors also in Ref. 17. Does that not imply that the impurity spin is overscreened by both source and drain conduction electrons down to $T=0$, since J is diagonal in the source/drain basis? If so, the argument that conductance is suppressed since Kondo screening only occurs with either source or drain, does not really seem to hold.

On the other hand the left panels of Fig. 2 clearly show that for small J_{sd} at low energies/temperatures the spin is only screened by either source or drain (thus supporting the interpretation of Kondo blockade proposed by the authors), although one can see that the screening contribution with the other channel comes closer to zero, when decreasing J_{sd} . So it seems that for small but finite J_{sd} the argument of the Kondo blockade being due Kondo screening just with one of the leads, actually holds, but for J_{sd} exactly zero the argument does not seem to hold. The authors should clarify this. Maybe an additional panel in Fig. 2 with J_{sd} exactly zero would help.

I think it would also be nice, in addition to Fig. 4, to see a plot of conductance versus J_{sd} in the vicinity of a $J_{sd}=0$ node, or conductance versus temperature in the vicinity of the J_{sd} node.

2) I understand that T^* is the temperature/energy scale for the RG flow from the intermediate overscreened NFL fixed point to the low temperature FL fixed point. Where does the expression $T^* \sim D(\rho_0 \delta)^2$; come from? In the case where the system passes through the NFL fixed point (small J_{sd}) it seems from the discussion that T_{Ke} , which is the Kondo temperature associated with the even screening channel, is identical or related with the two-channel temperature scale T_{2CK} (as defined e.g. in Ref. 17). Is that so and why?

3) Although I like the quite abstract discussion of the properties of an off resonant molecular junctions in terms of the 2CK model, I miss a bit the connection to the "physical reality" within the junction. For example, where does the different behavior of the two discussed junction types actually stem from? How can the different "Kondo blocking" and "Kondo effect" behaviors of

both junctions be understood e. g. in terms of actual electronic interference on the molecule? Where (in which molecular orbital) does the spin reside in the junction? Could one construct a multi-orbital Anderson model from molecular orbitals that would then give rise to the two-channel Kondo model in the off resonance limit?

4) I completely agree with the authors about the need to go beyond the simple (one-level) Anderson model paradigm for the description of correlation effects in molecular junctions. But I find the claim of generality of the proposed two-channel Kondo model a bit of an exaggeration. For example the effect of charge fluctuations, neglected in Kondo models by construction, could be very important in real molecular junctions. A multi-orbital Anderson model would be more appropriate then. It is also not really true that so far understanding of correlation effects in molecular junctions is entirely based on simple one-level Anderson model calculations. In fact multi-orbital Anderson models for molecular junctions have been constructed based on ab initio DFT calculations in a number previous works [R. Korytar and N. Lorente, *J. Phys.: Condens. Matter* 23, 355009 (2011); P. P. Baruselli et al., *Phys. Rev. B* 88, 245426 (2013); D. Jacob et al., *Phys. Rev. B* 88, 134417 (2013); S. Karan et al., *Physical Review Letters* 115, 016802 (2015)]

5) The interplay between quantum interference and Kondo effect is also important for the appearance of Fano lineshapes in conductance spectroscopy of adatoms, molecular junctions and quantum dots [A. Schiller and S. Hershfield, *Phys. Rev. B* 61, 9036 (2000); O. Ujsaghy et al., *Phys. Rev. Lett.* 85, 2557 (2000); R. Zitko, *Phys. Rev. B* 81, 115316 (2010)]. Maybe the authors could point out the difference and/or similarities between the interference effects leading to Fano lineshapes and the interference effects they are referring to.

In summary, despite my above criticism, I quite like the paper. The idea of quantum interference and Kondo effect in molecular junctions leading to multi-channel Kondo phenomena is novel, as far as I know, and may stimulate further theoretical as well as experimental research, searching for signatures of 2CK physics in molecular junctions. In particular the prediction of "Kondo blockade" for certain molecular junction could indeed be a very important result, provided the authors can clarify my confusion about it (Comment #1).

Reviewer #3 (Remarks to the Author)

The authors present a theoretical study of the interplay between Kondo physics and quantum interference (QI) in the context of molecular junctions. In particular, the authors predict that this interplay can lead, under certain circumstances, to the complete blockade of the electrical current at low bias. Moreover, they show that by playing with a gate voltage the system can be tuned from a regime where standard spin-1/2 Kondo physics shows up to a case where the low-bias current is completely suppressed. These conclusions are obtained with rigorous field-theoretical arguments supported by numerical renormalization group (NRG) calculations.

The concept of a QI-induced Kondo blockade put forward in this work is a fresh idea that deserves the attention of the communities of molecular electronics and mesoscopic physics. Moreover, the authors make a real effort to connect the minimal model for the discussion of this effect with more realistic descriptions of molecular junctions. Additionally, the conclusions are clearly backed up by well-established many-body theoretical methods. In this sense, I think that this work deserves publication and the visibility that a high-profile journal like Nature Communications offers. In any case, I think that the authors should improve certain aspects of the manuscript because it is not easy to follow some of arguments. Moreover, sometimes it is not clear the relevance or generality of the arguments and the text leaves the reader with some open questions. Below, I describe some of the changes that could be introduced to improve the manuscript.

1) From the analysis of the two (more realistic) examples of molecular junctions one gets the impression that whenever QI effects are expected (in the standard sense of non-interacting molecular junction models), one should be able to observe the QI-induced Kondo blockade. Is this actually so? What are the minimum requirements for QI to drastically modify the Kondo physics? Why is the behaviour of the two molecular junctions considered in this work qualitatively different? This is not clear from the text.

2) It would be desirable to provide some actual numbers for the relevant energy scales in these two examples. For instance, experimentalists may like to know the order of magnitude of the Kondo temperature in these systems, given some realistic estimates for the strength of the metal-molecule coupling and on-site Coulomb repulsion in the molecule.

3) The authors have focused on predictions for the conductance, but since they are using NRG they also have access to the information of the local density of states. This information is very important (although not enough) to give a first impression about the expectations at finite bias and the corresponding line shapes could be compared with the non-interaction models. Are these line shapes very different from Fano resonances or other QI line shapes? This could, at least, be mentioned in the text.

4) From a more technical point of view, as far as I understand the authors map the system Hamiltonian into an effective Hamiltonian with dimension equal to 2. However, a molecule can have in general a larger number of relevant energy states. From the manuscript one gets the impression that any model of a molecular junction can be reduced to the form of Eqs. 2-3 for the metal-molecule coupling. This mapping must have obvious limitations that are not clear from the text. So, the question that they author should clarify is under which conditions a molecular junction featuring QI can be described by an effective Hamiltonian of the form of H_{ex} in Eq. 2. In other words, is it possible to describe any type of QI in terms of this type of coupling

Hamiltonian?

Reviewer #4 (Remarks to the Author)

Questions and comments:

1. In the abstract the authors wrote “An exact framework is developed”, however, their formula is based on a 2nd-order Schrieffer-Wolff transformation, which means an effective theory. Could the authors explain what does the “exact” mean?
2. The author’s theory is an effective theory, which neglecting “unimportant” terms. So in principle, they cannot bring new phenomenon. Is there any experiment evidence or more advanced theory to prove the existence of Kondo blockade?
3. The author emphasize the multi-orbital structures of the molecule, however, I cannot understand how this gives the influence to the final result because in the derivation, the author does not sum the index of orbitals. Is it possible for the authors to give an explanation on this point in the real molecule part of the paper?
4. In page 2 line 106 left column, the author does not use summation on index ‘ i ’, but in line 112, they use a summation on index ‘ i ’, why?

This paper clarifies the possibility of Kondo blockade which is ignored in previous investigations. However, I am not familiar with the theoretical details. It may be published in Nat Comm after revision.

Reply to Referee #1:

We thank the Referee for a very thorough assessment of our manuscript, and for his/her view that the work is a *'sound and serious study'* that is *'clearly written and well-structured'*.

The Referee makes several insightful and helpful comments to consider *'before proceeding with publication'*, and we address these in full below.

1. *I believe the whole study is devoted to spin -1/2 molecules. This is explicitly stated in "the molecule typically hosts a spin-1/2 degree of freedom". However, from this statement one may infer that the analysis is also valid for larger-spin molecules, which is not necessarily true. In particular, for systems with $S > 1/2$, one could expect underscreened or fully-screened Kondo effects to emerge. The physics then may be completely different. This should be emphasized in the paper.*

The Referee is indeed correct that we considered only the spin-1/2 case in the present work. This situation might be considered typical -- for example, all-organic molecules in their 'natural' state are almost always spinless (all electrons paired), but can be charged by adding an electron by application of a back-gate voltage. The resulting radical with an odd number of electrons then often hosts a net spin-1/2. However we do agree that higher spins could also emerge (for example via Hund's rule type interactions or when dealing with metal complexes), and then interesting new physics may arise. We did not intend to imply that the same *physics* arises for higher spins, only that similar mappings to effective models can be performed in those cases.

To address this, we have replaced the word 'typically' with 'often' and added a new footnote [52] on higher spins. We also state in section *'Models and Mappings'*: "Higher-spin molecules can be treated similarly, but are not considered here." We now elaborate on the generalization to higher spins in the supplementary material.

2. *I think it may be worthwhile to introduce the notion of chain-like configuration (Wilson chain), when discussing the illustrations and Kondo screening clouds in Fig.2. Since from the Hamiltonian, Eq. (1), one cannot guess the chain configuration shown in Fig. 2, this will improve readability.*

We agree that the 1D chain structure illustrated in Fig. 2 merits further explanation in the text. However, the discussion in terms of Kondo clouds does not depend on the chains being specifically *Wilson chains* (in which case the mapping additionally involves a logarithmic discretization step). We depict the real-space physics in 1D for simplicity and visual clarity. Also, this is the case considered in Ref. 17, where a direct correspondence is made between the T-matrix (left panels of Fig. 2) and real-space physics in 1D (illustrated in the right panels). This connection is now made clear in the text.

To avoid an overly technical or cumbersome discussion in the main paper, we simply comment in the caption of Fig. 2 that the leads are depicted as 1D quantum wires, and that the Kondo clouds are real-space regions of high entanglement between the molecule and the leads. In section *'Emergent decoupling'*, we now elaborate on the real-space physics and comment explicitly on the 1D chain construction (see also new footnote [55]).

3. *The "Kondo blockade" results from the suppression of T-matrix when the coupling to one of the channels wins. The full energy-dependence of the T-matrix for 2CK was calculated in Phys. Rev. B 76, 155318 (2007). Also the frequency-dependent conductance, Eq. (29) in Suppl. Material, was derived in this paper. It would be good to add a reference to this paper.*

We have now added a citation to this relevant paper.

It should be noted, however, that the geometry of the two systems is crucially different. In molecular junctions, one has a strictly two-terminal setup and the conductance is serial -- i.e., from source to drain through the molecule, mediated by 'off-diagonal' cotunneling or exchange-cotunneling terms that are actually absent in the quantum dot setup. In the quantum dot device studied in the mentioned paper, the transport experiment is through a 'split lead'. This produces important differences: for example, at the 2CK fixed point, there is zero conductance through a molecular junction, despite finite t-matrix in the source or drain basis (the scattering S-matrix is zero). In this geometry, conductance may not simply be obtained from the t-matrix (except at $T=0$), as now highlighted in the main text. Also, it is the potential-scattering (cotunneling) that is Kondo blocked in molecular junctions; this cotunneling mechanism is absent anyway in the quantum dot model.

4. *In Fig. 5(c) and (e): at what T was the conductance calculated? There is a typo in the caption: Kondo blockade develops for $V_g = 2.625\text{eV}$ and the resonance for $V_g = 2.4\text{eV}$. Please make sure that the units for V_g are all the same: in axis label it is in V, in legend to (a) and (b) it is in eV.*

The conductance in Fig. 5(c) and (e) was calculated at $T=0$; however the results apply for all $T \ll T_K$. This information has now been added to the caption. We have corrected the typo and now use consistent units.

5. *It may be good to estimate T_K in Kelvins in Fig. 5: What is the temperature range to observe the Kondo blockade ($T < 0.1T_K$), is it experimentally accessible?*

The Kondo temperature T_K depends sensitively (exponentially) on the molecule-lead hybridization strength, and can therefore take a wide range of values. Furthermore, the effective 2CK parameters in general depend in a complicated way on the gate voltage V_g through the Schrieffer-Wolff transformation (which involves full diagonalization of the molecular moiety in the many-particle basis). To highlight generic aspects, we therefore plot data in Figs. 5(a) and (d) in terms of T/T_K . For the same reason, we chose parameters to avoid non-universal effects (T_K small compared with bare energy scales). We did not attempt an *ab initio* calculation of the Kondo temperature for specific molecules in this work.

However, it should be emphasized that the Kondo blockade arises on similar temperature scales to that of the standard Kondo effect in molecules. If one observes Kondo resonant conductance in a given system at experimentally-relevant temperatures, then the Kondo blockade should also be observable on tuning the gate to a QI node. In practice, we might expect to see Kondo blockade with associated T_K up to around 30K, which is not unusual for molecules.

The specific examples shown in Fig. 5 are conjugated hydrocarbons, and we used a PPP model to describe the pi system. In such cases, the effective bandwidth cutoff D of the resulting 2CK model is essentially set by the large charging energy of the molecule, which in turn is controlled by the Coulomb repulsion U . This is taken as 11eV within the standard Ohno parametrization (see supplementary material and Ref 64). In general, we estimate D to be roughly in the range 1-10eV.

At the Kondo blockade in Fig. 5, we then have a Kondo temperature $T_K \approx 10\text{K}$ in Fig. 5c, while for Fig. 5f it is around 0.1K. The molecule-lead hybridization here was chosen so that the Kondo temperatures fall roughly in the range observed in typical experiments on molecular junctions. We anticipate that signatures of Kondo blockade could be observable, even if the precise conductance node is not fully developed at experimental base temperatures.

We have now commented on these points in connection with Fig. 5.

6. *I do not understand the conductance enhancement in Fig. 5(e): is it due to the Kondo effect? I guess so, since this is the conductance for Coulomb blockade of spin $\frac{1}{2}$ system. However, at V_g where $G = 2e^2/h$, $T_K=0$, such that there is no Kondo effect. Could the Authors explain this behavior in more detail?*

In the case of Fig. 5 (e), the Kondo effect (involving both source and drain leads) is fully developed at $V_g=2.4$ V, and the corresponding conductance is maximal. The Kondo temperature T_K is however not zero, but just a finite minimum value as a function of V_g (we have now added a comment on this).

We note that this point is not precisely at a quantum interference node (see elaborated comment in first paragraph of section 'Gate-tunable QI in Kondo-active molecules' and Fig. 2 of the supplementary material). This is because exchange and potential scattering terms are typically both present in real systems, and conductance is affected by mutual renormalization of these terms.

7. *The conductance suppression due to the Kondo blockade (Fig. 5) looks very small, I wonder if it would be experimentally measurable at all. Can it be enhanced in other systems? Could the Authors comment on that?*

The Kondo blockade quenches the expected **perturbative** conductance obtained at higher temperatures. This is on the order of $(\rho_0 W_{sd})^2$ and is typically small, given that the molecular junctions under consideration are off-resonance. The Kondo blockade produces a conductance node, $G(0)=0$; on the scale of the perturbative expectation, the effect is dramatic. However, the conductance enhancement due to the Kondo effect is comparatively very large (of order e^2/h) and may indeed dwarf the cotunneling conductance or Kondo blockade signature.

The conductance node associated with Kondo Blockade might therefore be most cleanly observed in situations where there is no nearby Kondo resonance, so that the conductance 'contrast' is maximized. We also note that W_{sd} is itself proportional to the molecule-lead tunneling matrix elements, and therefore the high- T conductance is enhanced in junctions where this coupling is stronger. We have added a comment on this in the 'Kondo blockade' section.

8. *When referring to NRG method, I encourage the Authors to cite also the original Wilson's paper.*

We agree, and have now added the original reference.

9. *A similar conductance suppression due to quantum interference was also observed in double quantum dot systems, see e.g. Phys. Rev. Lett. 87, 216601(2001), Phys. Rev. Lett. 103, 266806 (2009), Phys. Rev. B 81, 115316 (2010). The conductance suppression can be understood there by invoking the two-stage Kondo effect and is therefore also an example of strong-correlation-mediated current blockade.*

We have added these references as good examples of how electronic interactions can strongly modify conductance in nanostructures.

We note however that the conductance suppression in those cases is a consequence of having two spin-1/2 impurities in a single-channel setup (or a spin-1 configuration with two channels). Neither situation relates directly to the physics discussed in our paper.

The two-stage Kondo effect in the side-coupled geometry is a special case in which the successive screening of the two impurities confers first the regular Kondo $\pi/2$ phase shift, and then a second $\pi/2$ phase shift to the conduction electrons – a situation continuously connected to the inter-impurity singlet regime of zero $T=0$ conductance. The Hund's rule spin-1 case is not considered in the present work (as per our answer to question 1). The conductance suppression in these cases arise from completely different mechanisms to the Kondo blockade, which depends instead on quantum interference and should be accessible in molecular junctions by tuning gate voltages.

Reply to referee #2:

We thank the Referee for his/her careful and detailed review of our manuscript. In particular, we are grateful for their view that the work ‘*may stimulate further theoretical as well as experimental research*’ and that the predicted Kondo blockade could be ‘*a very important result*’. The Referee makes a number of interesting and useful comments which we address below. Especially, we wish to clarify the matter of the first question:

1. *The "Kondo blockade" occurs at an interference node in the source-drain exchange coupling ($J_{sd}=0$). As far as I understand, in this limit the model should be governed by two-channel Kondo physics down to zero temperature, leading to frustration due to the competition between even and odd screening channels, as explained by one of the authors also in Ref. 17. Does that not imply that the impurity spin is overscreened by both source and drain conduction electrons down to $T=0$, since J is diagonal in the source/drain basis? If so, the argument that conductance is suppressed since Kondo screening only occurs with either source or drain, does not really seem to hold.*

On the other hand the left panels of Fig. 2 clearly show that for small J_{sd} at low energies/temperatures the spin is only screened by either source or drain (thus supporting the interpretation of Kondo blockade proposed by the authors), although one can see that the screening contribution with the other channel comes closer to zero, when decreasing J_{sd} . So it seems that for small but finite J_{sd} the argument of the Kondo blockade being due Kondo screening just with one of the leads, actually holds, but for J_{sd} exactly zero the argument does not seem to hold. The authors should clarify this. Maybe an additional panel in Fig. 2 with J_{sd} exactly zero would help.

The Kondo blockade phenomenon we discuss indeed arises at an exchange cotunneling node $J_{sd}=0$, whence the effective model is the regular 2CK model, plus additional ‘direct’ cotunneling terms (off-diagonal potential scattering W_{sd}). The latter are responsible for the finite (perturbative) conductance at higher temperatures, which get quenched at lower temperature $T \ll T_K$ due to emergent Kondo physics. However, the situation referred to by the Referee, where the molecule spin-1/2 is *overscreened*, is a special case where the source and drain leads are symmetrically coupled, $J_{ss}=J_{dd}$. In this case, the frustration of Kondo screening between the two channels exists down to $T=0$ (that is, the Fermi liquid scale $T^*=0$). In the cartoons of Fig. 2, the overscreening Kondo cloud size would then diverge.

We did not discuss this particular scenario in the paper, because it is rather artificial: in any real system, there will inevitably be some asymmetry between source and drain (however small that might be). This always relieves the frustration below the scale T^* , which in general is finite. For $T \ll T^*$, screening by one or other of the channels ‘wins’ and then one sees the Kondo blockade. This point is now clarified and also further discussed in the supplementary material, section S6.

Fig. 2, and the surrounding discussion, relates to the case where J_{ss} and J_{dd} are **not** precisely equal: this is quantified by the J_- parameter, which is small but finite in the calculations presented (see updated Fig. 2 caption). We have now emphasized in the text that J_- is finite and why this is the physically relevant scenario of interest for molecular junctions. See also footnote [61] which further mentions that for when $J_{ss}=J_{dd}$, the Kondo blockade effect vanishes.

I think it would also be nice, in addition to Fig. 4, to see a plot of conductance versus J_{sd} in the vicinity of a $J_{sd}=0$ node, or conductance versus temperature in the vicinity of the J_{sd} node.

In Fig. 4 we plotted the conductance versus temperature when *precisely* at a node in J_{sd} . This behavior is generic, since $G(T)/G_{pert}$ versus T/T_K is a universal curve, as now mentioned explicitly (G_{pert} is the high- T perturbative result proportional to W_{sd}^2). We also expect that this behavior could be experimentally observable by tuning to a quantum interference node in J_{sd} using gate voltages (and provided the Kondo temperature is sufficiently large compared with the experimental base temperature that the blockade is well-developed).

However, when W_{sd} and J_{sd} are *both* finite, the resulting physics is complicated and not universal due to mixing of the channels and mutual renormalization from cross terms in the conductance proportional to $W_{sd}J_{sd}$ (see comment in first paragraph of section ‘Gate-tunable QI in Kondo-active molecules’ and Fig. 2 of the supplementary material). Furthermore, J_{sd} is not independently and directly tunable: all of the effective 2CK parameters are modified on tuning gate voltages, and this happens in a complicated way that depends on the particular molecular structure. Therefore the evolution of conductance with J_{sd} in the *vicinity* of a node is not as such a robust experimental observable. Rather, it is the physics *at* the nodes that is generic (we speculate that the nodes may be topological). Examples of how the conductance evolves with *gate* near QI nodes are already shown in Fig. 5.

2. *I understand that T^* is the temperature/energy scale for the RG flow from the intermediate overscreened NFL fixed point to the low temperature FL fixed point. Where does the expression $T^* \sim D(\rho_0\delta)^2$; come from? In the case where the system passes through the NFL fixed point (small J_{sd}) it seems from the discussion that T_K^e , which is the Kondo temperature associated with the even screening channel, is identical or related with the two-channel temperature scale T_K^{2CK} (as defined e.g. in Ref. 17). Is that so and why?*

The quadratic dependence of the Fermi liquid scale T^* on RG relevant perturbations δ (such as channel asymmetry) is known from the conformal field theory analysis of the 2CK problem by Affleck and Ludwig. A reference has now been added. This applies in the regime where perturbations are small such that there is scale separation $T^* \ll T_K$, where T_K is the flow to the 2CK critical point and T^* is the flow away from it. The result is basically a consequence of the Majorana structure of the 2CK critical point and the scaling dimension $\frac{1}{2}$ operators which can be associated with the relevant perturbations. For very small δ , we have $J_e \approx J_o$ and so $T_K^e \approx T_K^o \approx T_K^{2CK}$. On the other hand, T^* is controlled by δ , which only sets in at much lower temperatures. This discussion has now been added.

For larger perturbation strength δ , the 2CK critical physics is not probed -- RG flow proceeds directly from the local moment fixed point to the Kondo strong coupling fixed point. The low-energy physics is again Fermi-liquid-like, and indeed it is all continuously connected for any δ (of the same sign). Here the Fermi liquid scale is just the Kondo temperature, which is set by the larger of the two exchange couplings J_e . We denote this Kondo scale T_K^e .

3. *Although I like the quite abstract discussion of the properties of off-resonant molecular junctions in terms of the 2CK model, I miss a bit the connection to the "physical reality" within the junction. For example, where does the different behavior of the two discussed junction types actually stem from? How can the different "Kondo blocking" and "Kondo effect" behaviors of both junctions be understood e. g. in terms of actual electronic interference on the molecule? Where (in which molecular orbital) does the spin reside in the junction? Could one construct a multi-orbital Anderson model from molecular orbitals that would then give rise to the two-channel Kondo model in the off resonance limit?*

As appreciated by the Referee, the power of our effective theory is that one does not have to worry about where the spin resides or the character of the molecular orbitals producing the quantum interference. All one needs is the effective 2CK parameters; and indeed one can study the various possibilities arising in this effective model independently of any specific realization.

However, the physical information mentioned by the Referee can still be extracted from the calculation. Specifically, it is contained in the details of the 2CK mapping discussed in the supplementary material, section S1. Because the molecule is *interacting*, one has to work in the many-particle basis; the generalization of the standard (non-interacting) molecular orbitals to the interacting case are the so-called Feynman-Dyson orbitals. These are tunneling matrix elements between different molecular wavefunctions and are therefore spread over the entire molecule. Quantum interference effects can be

seen as depletion of density in these Feynman-Dyson orbitals. In general the spin wavefunction spreads over the whole molecule, although it may be localized more in some regions than others, depending on details of the bonding. These issues were addressed in our earlier PRB, Ref. [45]. We have now added a comment on these points in section ‘Models and mappings’.

In the case of conjugated organic molecules, the relevant multi-orbital Anderson model can be obtained using the PPP model (extended Hubbard model) for the pi system. As mentioned in the text, this was our starting point for the 2CK mapping for the molecules considered in Fig. 5. In principle, the PPP model could also be directly implemented in NRG (and this is something we are working on). However, one is then restricted to comparatively small systems, and the physical content is arguably obfuscated rather than clarified by the additional complexity.

The differences between the two molecules considered in Fig. 5 relate to the number and position of the nodes in the effective 2CK parameters shown in the supplementary material (Figs. S1 and S2). For example, both molecules show Kondo blockade due to a node in J_{sd} , but this arises at $V_g=0$ for the benzyl radical whereas there are two nodes at \pm finite V_g for the isoprene-like molecule. This can be traced back to the *symmetry* of the molecule and contacting geometry. The QI nodal properties were connected to the molecular structure in our earlier PRB Ref. [45] for alternant molecules (conjugated hydrocarbons with a bipartite pi system). In the case of odd-membered alternant molecules such as those in Fig. 5, J_{sd} must have an odd number of nodes if the source and drain electrodes connect to sites on different sublattices (e.g. the benzyl radical, Fig. 5a), but an even number of nodes if the contacts are on the same sublattice (e.g. the isoprene-like molecule, Fig. 5b). Similar conditions can be derived for the nodal structure of the other 2CK parameters.

Note, however, that the Kondo blockade phenomenon is not restricted to odd-alternant molecules: any system exhibiting a QI node in the exchange cotunneling will exhibit a Kondo blockade.

The strong Kondo resonance arising for the isoprene-like molecule arises because $J_{ss} \approx J_{dd}$ (see Eq. 5) which is a consequence of the parity symmetry with respect to the contacting geometry. By contrast, there is no such symmetry for the benzyl molecule and J_{dd} happens to dominate.

We have now provided an explanation of this in section ‘Gate-tunable QI in Kondo-active molecules’.

4. *I completely agree with the authors about the need to go beyond the simple (one-level) Anderson model paradigm for the description of correlation effects in molecular junctions. But I find the claim of generality of the proposed two-channel Kondo model a bit of an exaggeration. For example the effect of charge fluctuations, neglected in Kondo models by construction, could be very important in real molecular junctions. A multi-orbital Anderson model would be more appropriate then.*

The present paper focuses on off-resonance molecular junctions with a net spin-1/2.

Deep within a Coulomb blockade diamond, where we perform our exact mapping to the effective low-energy two-channel Kondo model, charge fluctuations are indeed frozen out. Put more precisely: going beyond the second-order Schrieffer-Wolff transformation presented in the supplementary material (section S1) does give corrections from virtual transitions involving higher charge states, but these are formally *RG irrelevant*. They do not affect the underlying physics – they can only modify the effective energy scales appearing in the problem. Indeed, these corrections are small anyway, being suppressed by the large molecule charging energy. This kind of effect is well-known already when comparing the single-impurity Anderson model and the single-impurity Kondo model, which have identical low-energy physics for all $T_K \ll U$. A comment on this has now been made in the section ‘Models and mappings’.

Needless to say, however, the off-resonance molecular junctions considered here are not the most general scenario. The Referee is certainly correct that interesting physics will arise in the charge-fluctuation regime in the vicinity of a step in the Coulomb blockade staircase (in fact, this is the topic of

Ref. 29). These effects are beyond the scope of the present paper. We have however now drawn the reader's attention to this possibility in footnote [51].

We believe that we have made clear the regime of applicability of the effective 2CK model. In the introduction: "we construct an effective model describing off-resonant conductance". In section 'Models and mappings': "Deep in a Coulomb diamond, a substantial charging energy, E_C , must be overcome to either add or remove an electron. Provided $\Gamma \ll E_C$ the Hamiltonian can therefore be projected onto the subspace with a fixed number of electrons on the molecule."

It is also not really true that so far understanding of correlation effects in molecular junctions is entirely based on simple one-level Anderson model calculations. In fact multi-orbital Anderson models for molecular junctions have been constructed based on ab initio DFT calculations in a number of previous works [R. Korytar and N. Lorente, J. Phys.: Condens. Matter 23, 355009 (2011); P. P. Baruselli et al., Phys. Rev. B 88, 245426 (2013); D. Jacob et al., Phys. Rev. B 88, 134417 (2013); S. Karan et al., Physical Review Letters 115, 016802 (2015)]

We agree that other important work has been done in going beyond the single-orbital paradigm, and we have now cited the mentioned papers as good examples of this. The goal of our paper is rather to formulate a simple but general theoretical framework describing off-resonant spin-1/2 molecular junctions that allows multi-orbital effects to be understood in strongly interacting systems. The power of our approach is demonstrated by our prediction of a novel effect, the Kondo blockade.

5. *The interplay between quantum interference and Kondo effect is also important for the appearance of Fano lineshapes in conductance spectroscopy of adatoms, molecular junctions and quantum dots [A. Schiller and S. Hershfield, Phys. Rev. B 61, 9036 (2000); O. Ujsaghy et al., Phys. Rev. Lett. 85, 2557 (2000); R. Zitko, Phys. Rev. B 81, 115316 (2010)]. Maybe the authors could point out the difference and/or similarities between the interference effects leading to Fano lineshapes and the interference effects they are referring to.*

In this context of transport, the Fano effect can result from the combination of quantum interference and Kondo effect – although in fact Kondo physics was not an ingredient in Fano's original setting and strong correlations are not essential for Fano. The Fano effect is not related to the present discussion of the Kondo blockade. We anticipated that readers might wonder about the relation to the Fano effect, and so the existing supplementary material already contains a couple of paragraphs discussing this issue (see end of section S6). We have endeavored to make this clearer in the new manuscript, and have flagged up this discussion in the text of the main paper. We now also cite the mentioned papers.

The key difference is that the Fano lineshape can be completely understood in terms of a *single-channel problem featuring a modified hybridization function*. The quantum interference is entirely on the level of the non-interacting effective conduction electron bath, which consists of both substrate and STM tip, and whose combined density of states is modified locally at the impurity by the direct tip-substrate tunneling. By contrast, the quantum interference we discuss in relation to molecular junctions arises from tunneling **through the interacting molecule itself**, and the problem remains **irreducibly two-channel** in character.

The 2010 paper by R. Zitko additionally considers a side-coupled two-impurity scenario which is also nontrivial. However, it is not relevant to the present discussion, since the setup does not involve a net spin-1/2 and it is effectively single-channel. The conductance suppression there does not have its origin in quantum interference but rather the two-stage Kondo effect for successive screening of two spin-1/2 impurities. The precise conductance node in that case is also an artifact of the symmetry of the setup (the two dots must be identical otherwise the two phase shifts do not precisely cancel).

Reply to referee #3:

We thank the referee for a thorough and detailed assessment of our manuscript. We are very happy that the Referee agrees that our findings constitute ‘*a fresh idea that deserves the attention of the communities of molecular electronics and mesoscopic physics*’ and that the work ‘*deserves publication and the visibility that a high-profile journal like Nature Communications offers*’. Furthermore, we are grateful for the constructive comments raised by the Referee, which we address below.

1. *From the analysis of the two (more realistic) examples of molecular junctions one gets the impression that whenever QI effects are expected (in the standard sense of non-interacting molecular junction models), one should be able to observe the QI-induced Kondo blockade. Is this actually so? What are the minimum requirements for QI to drastically modify the Kondo physics? Why is the behaviour of the two molecular junctions considered in this work qualitatively different? This is not clear from the text.*

The Kondo effect in molecular junctions is dramatically modified by quantum interference when it produces a node in the *exchange* cotunneling, $J_{sd}=0$. Rather than the usual Kondo-enhanced conductance below T_K , we then see Kondo-**blocked** conductance.

However, the QI which leads to this node is not related to that arising in non-interacting models – there is no molecular-localized spin without interactions and hence no exchange coupling. The Kondo physics depends on this, and the QI affecting it is similarly an interaction-based phenomenon.

However, another important prediction is that there are two different types of QI in interacting systems – the other produces a node in the spin-independent cotunneling (potential scattering), $W_{sd}=0$. For molecules with small charging energies (weak interactions), this cotunneling is perturbatively connected to the non-interacting case, and therefore one can deduce the cotunneling QI using perturbation theory. In this case, the spin-independent cotunneling can be related to the ‘standard’ QI arising from non-interacting molecular orbitals, and depends similarly on molecular structure, contacting geometry and gate voltages. We have now added a discussion in the section ‘*Models and mappings*’.

The minimum requirement for Kondo blockade is an off-resonant molecular junction with net spin-1/2, intramolecular interactions, and sufficient orbital complexity to produce a QI node in the exchange cotunneling, J_{sd} . The QI nodal structure can be determined from the molecular structure, symmetry, and contacting geometry. We have now highlighted this in the revised manuscript.

The differences between the two molecules considered in Fig. 5 relate to the number and position of the nodes in the effective 2CK parameters shown in the supplementary material (Figs. S1 and S2). For example, both molecules show Kondo blockade due to a node in J_{sd} , but this arises at $V_g=0$ for the benzyl radical whereas there are two nodes at \pm finite V_g for the isoprene-like molecule. This can be traced back to the *symmetry* of the molecule and contacting geometry. The QI nodal properties were connected to the molecular structure in our earlier PRB Ref. [45] for alternant molecules (conjugated hydrocarbons with a bipartite pi system). In the case of odd-membered alternant molecules such as those in Fig. 5, J_{sd} must have an odd number of nodes if the source and drain electrodes connect to sites on different sublattices (e.g. the benzyl radical, Fig. 5a), but an even number of nodes if the contacts are on the same sublattice (e.g. the isoprene-like molecule, Fig. 5b). Similar conditions can be derived for the nodal structure of the other 2CK parameters.

Note, however, that the Kondo blockade phenomenon is not restricted to odd-alternant molecules: any system exhibiting a QI node in the exchange cotunneling will exhibit a Kondo blockade.

The strong Kondo resonance arising for the isoprene-like molecule arises because $J_{ss} \approx J_{dd}$ (see Eq. 5) which is a consequence of the parity symmetry with respect to the contacting geometry. By contrast, there is no such symmetry for the benzyl molecule and J_{dd} happens to dominate.

We have now provided an explanation of this in section ‘Gate-tunable QI in Kondo-active molecules’.

2. *It would be desirable to provide some actual numbers for the relevant energy scales in these two examples. For instance, experimentalists may like to know the order of magnitude of the Kondo temperature in these systems, given some realistic estimates for the strength of the metal-molecule coupling and on-site Coulomb repulsion in the molecule.*

The Kondo temperature T_K depends sensitively (exponentially) on the molecule-lead hybridization strength, and can therefore take a wide range of values. Furthermore, the effective 2CK parameters in general depend in a complicated way on the gate voltage V_g through the Schrieffer-Wolff transformation (which involves full diagonalization of the molecular moiety in the many-particle basis). To highlight generic aspects, we therefore plot data in Figs. 5(a) and (d) in terms of T/T_K . For the same reason, we chose parameters to avoid non-universal effects (T_K small compared with bare energy scales). We did not attempt an *ab initio* calculation of the Kondo temperature for specific molecules in this work.

However, it should be emphasized that the Kondo blockade arises on similar temperature scales to that of the standard Kondo effect in molecules. If one observes Kondo resonant conductance in a given system at experimentally-relevant temperatures, then the Kondo blockade should also be observable on tuning the gate to a QI node. In practice, we might expect to see Kondo blockade with associated T_K up to around 30K, which is not unusual for molecules.

The specific examples shown in Fig. 5 are conjugated hydrocarbons, and we used a PPP model to describe the pi system. In such cases, the effective bandwidth cutoff D of the resulting 2CK model is essentially set by the large charging energy of the molecule, which in turn is controlled by the Coulomb repulsion U . This is taken as 11eV within the standard Ohno parametrization (see supplementary material and Ref 64). In general, we estimate D to be roughly in the range 1-10eV.

At the Kondo blockade in Fig. 5, we then have a Kondo temperature $T_K \approx 10K$ in Fig. 5c, while for Fig. 5f it is around 0.1K. The molecule-lead hybridization here was chosen so that the Kondo temperatures fall roughly in the range observed in typical experiments on molecular junctions. We anticipate that signatures of Kondo blockade could be observable, even if the precise conductance node is not fully developed at experimental base temperatures.

We now discuss the above issue of Kondo temperatures in connection with Fig. 5.

3. *The authors have focused on predictions for the conductance, but since they are using NRG they also have access to the information of the local density of states. This information is very important (although not enough) to give a first impression about the expectations at finite bias and the corresponding line shapes could be compared with the non-interaction models. Are these line shapes very different from Fano resonances or other QI line shapes? This could, at least, be mentioned in the text.*

The local electronic density of states of the leads at the junction is proportional to the imaginary part of the T-matrix, plotted in the source-drain basis in the left-hand panels of Fig. 2. As the Referee points out, one can understand a lot about the physics and expected conductance from this information. Indeed, the Kondo blockade is shown to be related to a local suppression in the lead density of states at the Fermi level for $T \ll T_K$ (although a more sophisticated treatment is required to obtain the full temperature

dependence). We have now made a comment in connection with Fig. 2, emphasizing the relationship with the density of states.

With NRG we are confined to linear response conductance. However, finite bias is addressed analytically in the quantum critical regime in Eq. 7.

Quantum interference phenomena in non-interacting models are rather different (there is no universality in terms of T_K , for example). The Fano effect is unrelated to the conductance suppression discussed here, and different (asymmetric) lineshapes are obtained in that case (see supplementary material section S6 for a discussion of this). A discussion has now been added in the main text of the paper.

- 4. From a more technical point of view, as far as I understand the authors map the system Hamiltonian into an effective Hamiltonian with dimension equal to 2. However, a molecule can have in general a larger number of relevant energy states. From the manuscript one gets the impression that any model of a molecular junction can be reduced to the form of Eqs. 2-3 for the metal-molecule coupling. This mapping must have obvious limitations that are not clear from the text. So, the question that they author should clarify is under which conditions a molecular junction featuring QI can be described by an effective Hamiltonian of the form of H_{ex} in Eq. 2. In other words, is it possible to describe any type of QI in terms of this type of coupling Hamiltonian?*

The key conditions for applicability of the effective 2CK model (Eqs 2-3) are that the molecule be ‘off-resonant’ and hosts a net spin-1/2. If these conditions are satisfied, the 2CK mapping described in the supplementary material S1 is valid. These conditions are stated in the introduction of the paper and in the section ‘Models and mappings’ (and more stringently defined in the supplementary material).

The physics of the full system is accurately described by the effective model at temperatures and tunnel couplings much less than the molecule charging energy, $T, \Gamma \ll E_C, U$.

In practice, this scenario is quite ubiquitous: off-resonant means simply an interacting molecule deep within a Coulomb diamond, such that the number of electrons on the molecules is essentially fixed and charge fluctuations are suppressed by the large molecule charging energy. And even though most molecules in their ‘natural’ state are spinless (all electrons paired in molecular orbitals), the molecule can be charged in a molecular junction context using the back-gate voltage to add or remove an electron. The resulting species then usually carries a spin-1/2 (although it may be delocalized). This is now mentioned in footnote [52]. We now also mention that higher spin molecules are not studied here.

In this sense the applicability of the effective model is very general, and all the microscopic details of real molecular junctions (including all QI effects) just end up in the effective 2CK parameters, Eq. 3. The reduced complexity of the effective model is the reason that we can make simple but general predictions about the resulting physics.

As mentioned in the conclusion, however, we did not account for vibrations or dissipation. Potentially these are a different source of quantum interference that we will consider in future work.

Reply to referee #4:

We thank the referee for a thorough assessment of our manuscript. We were especially grateful for his/her view that *'this paper clarifies the possibility of Kondo blockade which is ignored in previous investigations'*, and that *'It may be published in Nat Comm after revision.'*

1. *In the abstract the authors wrote "An exact framework is developed", however, their formula is based on a 2nd-order Schrieffer-Wolff transformation, which means an effective theory. Could the authors explain what does the "exact" mean?*

The Referee is correct that we derive and use an effective model for the molecular junctions. It is this effective 2CK model that we analyze analytically and numerically with NRG in the main part of the paper. The detailed derivation of the model is presented in the supplementary material, section S1. Our results on the effective model are exact.

The 2CK model is derived for molecules hosting a net spin-1/2 in the 'off-resonant' regime (deep within a Coulomb blockade diamond), as advertised in the introduction and in the section *'Models and mappings'* (a more stringent mathematical definition is given in the supplementary material).

The effective model is therefore valid at temperatures and tunnel couplings much less than the molecule charging energy $T, \Gamma \ll E_C, U$. In particular, physical quantities of the full molecular junction system at low temperatures can be accurately calculated using the effective model.

Our claim that the theoretical framework is 'exact' is based on the renormalization group concept. Although the effective model is derived perturbatively to second order in the molecule-lead coupling, the neglected corrections coming from higher-order perturbation theory are formally **RG irrelevant**. This means that as the energy or temperature scale decreases, they become less and less important and asymptotically vanish. For this reason, these neglected terms cannot affect the underlying physics of the molecular junction problem. However they can modify the effective energy scales emerging, such as the Kondo temperature T_K (although these effects are also expected to be small and parametrically suppressed by the large molecular charging energy). This is why we emphasize generic and universal features, rescaling in terms of T/T_K . We have added a discussion on this in section *'Models and mappings'*.

Note that the effective model, although simplified, is still non-trivial and is solved non-perturbatively.

2. *The author's theory is an effective theory, which neglecting "unimportant" terms. So in principle, they cannot bring new phenomenon. Is there any experiment evidence or more advanced theory to prove the existence of Kondo blockade?*

The power of the effective theory we develop is that it can be analyzed exactly and all of the different possible scenarios that can in principle arise in molecular junctions can be classified and assessed. The complexity of the microscopic model means that a brute-force approach (even if that were possible) would not yield any new conceptual understanding nor general predictive power (beyond the case-by-case basis). Our surprising new result, elucidating the interplay between Kondo and quantum interference, had previously been missed.

However, the Referee is correct that the same Kondo blockade physics must and does arise in the original microscopic multi-orbital model. The low-temperature physics is identical, even if differences appear at higher temperatures where non-universal effects dominate (see above reply to Q1).

We understand from our experimental colleagues that experimental data for single-molecule junctions frequently display 'unconventional' behavior that cannot be explained within the usual single-orbital Anderson paradigm (especially as a function of gate voltage, see e.g. Ref. [28]). We hope that our paper will provide a guide to understanding these experiments and will stimulate the search for robust experimental signatures of the Kondo blockade.

3. *The author emphasize the multi-orbital structures of the molecule, however, I cannot understand how this gives the influence to the final result because in the derivation, the author does not sum the index of orbitals. Is it possible for the authors to give an explanation on this point in the real molecule part of the paper?*

The multi-orbital structure is in fact fully accounted for in the mapping to the effective 2CK model discussed in the supplementary material, section S1. We have tried to make this more transparent in the revised manuscript by emphasizing that the wavefunctions $|\psi_n\rangle$ are obtained by full diagonalization of the (isolated) molecule Hamiltonian in the many-particle basis. The basis from which they are constructed is the complete set of atomic or molecular orbitals in the microscopic formulation, and therefore in general $|\psi_n\rangle$ span the entire molecule and involve all orbitals. The spin of the effective Kondo model is obtained by projecting onto the pair of lowest energy degenerate eigenstates. The exchange couplings depend on the tunneling matrix elements between the leads and specific frontier orbitals of the molecule, as described in H_{hyb} . One must therefore evaluate the weight of these orbitals in $|\psi_n\rangle$ to compute the parameters in Eq. 3.

4. *In page 2 line 106 left column, the author does not use summation on index 'i', but in line 112, they use a summation on index 'i', why?*

Related to the reply to Q3 above, the part of the Hamiltonian coupling the molecule and the leads is specified by H_{hyb} . It involves a tunnel-coupling term between the orbital (or effective orbital) of the leads localized at the junction, and a specific frontier orbital of the molecule (in fact, H_{hyb} could serve as a definition of this frontier orbital). There is no sum over molecular orbitals here because the molecule-lead coupling is **local** (this is now emphasized in the revised manuscript). However, the gate voltage V_g shifts the energy levels of all molecular orbitals, and hence H_g does involve the sum over orbitals i . We have also added a comment in the paper to clarify this issue.

Summary of changes:

All changes are marked in red in the attached manuscript pdf

- Line 70: Added comment on many-body QI due to interactions
- Line 75: Effective model valid for spin-degenerate molecules
- Line 81: Connection with density of states
- Line 101 : Previous work beyond single AIM paradigm cited
- Line 116: Molecule-lead coupling is local
- Line 119: All MOs shifted by gate
- Line 125: New footnote 50 on charge fluctuation physics
- Line 126: Eigenstates of isolated molecule in many-particle basis
- Line 128: New footnote 51 on higher spin molecules
- Line 134: Higher spin molecules not treated
- Line 142: Explanation of the sense in which the 2CK model is exact
- Fig 2, Caption: Connection between T-matrix and density of states; source/drain asymmetry; elaboration of real-space interpretation and 1D chain structure.
- Line 169: Clarification of QI types for interacting systems and physical character of QI
- Line 221: Clarification of source/drain asymmetry
- Line 259: Explanation of why $T_K^e \approx T_K^{2CK}$ for small δ
- Line 265: Reference added for T^* dependence on δ and low- T frustration relief
- Line 275: Note on quantum critical point added
- Line 278: Explication of real-space physics and new footnote 55
- Line 305: Note on relation between conductance and T-matrix
- Line 400: Universal properties of Kondo blockade
- Line 404: Clean observation of Kondo blockade
- Line 412: Relation to Fano effect
- Line 432: Incorporated previous footnote 48 into main text
- Updated Fig. 5 with consistent labelling
- Line 476: $G(0)$ is calculated at $T=0$
- Line 487: Kondo temperature always finite
- Line 488: Note on Kondo temperatures in real systems
- Line 494: Note on Kondo temperatures in Fig. 5
- Line 509: Explanation of difference between two molecules in Fig. 5
- Line 530: Expectation for appearance of Kondo blockade
- Added Refs 18, 24, 26, 30, 33, 46-49, 54, 62, 63

Supplementary Material:

- Page 1: Clarification that the wavefunctions are in the many-particle basis
- Page 2: Note added on generalization to higher spin molecules
- Page 3: Details added on regime of validity for effective model
- Page 4: Note added on the advantage of the effective model over the microscopic model
- Page 11: Further details on relation to Fano effect
- Page 12: Note added on relation to two-stage Kondo in double dots
- Page 13: Details added on calculation of effective 2CK parameters

Reviewers' Comments:

Reviewer #1 (Remarks to the Author):

The authors have addressed all the comments raised in my Report and revised the manuscript accordingly. In my opinion, the paper is now ready for publication.

Reviewer #2 (Remarks to the Author):

The authors have answered all my questions and comments to my satisfaction. I think the revised version of the manuscript is clearer now in the few parts where I found it unclear. As I have said already in my previous report, I think that this is a nice paper that predicts a very interesting novel phenomenon, the Kondo blockade, to occur in single molecule electronic devices, and that will probably stimulate further experimental and theoretical research. I therefore strongly recommend publication of the revised manuscript in Nature Communications.

Reviewer #3 (Remarks to the Author):

The authors have made a big effort to address all the comments and questions that I raised in my previous report. Overall, their answers and arguments are quite satisfactory. Moreover, the new comments introduced in the manuscript have improved its readability and the paper now is more precise and accurate. So in this sense, I am happy to recommend its publication without further revision.